

# Using Neural Network Ensembles to Separate Biogeochemical and Physical Components in Earth System Models

Christopher Holder[1], Anand Gnanadesikan[1], Marie Aude-Pradal[1]

[1]Morton K. Blaustein Department of Earth and Planetary Sciences, Johns Hopkins University, Baltimore, MD, United States of America

Correspondence to: Christopher Holder (cholder2@jh.edu)

**Abstract.** Earth system models (ESMs) are useful tools for predicting and understanding past and future aspects of the climate system. However, the biological and physical parameters used in ESMs can have wide variations in their estimates. Even small changes in these parameters can yield unexpected results without a clear explanation of how a particular outcome was reached. The standard method for estimating ESM sensitivity is to compare spatiotemporal distributions of variables from different runs of a single ESM. However, a potential pitfall of this method is that ESM output could match observational patterns because of compensating errors. For example, if a model predicts overly weak upwelling and low nutrient concentrations, it may compensate for this by allowing phytoplankton to have a high sensitivity to nutrients. Recently, it has been demonstrated that neural network ensembles (NNEs) are capable of extracting relationships between predictor and target variables within ocean biogeochemical models. Being able to view the relationships between variables, along with spatiotemporal distributions, allows for a more mechanistically based examination of ESM outputs. Here, we investigated whether we could apply NNEs to help us determine why different ESMs produce different results. We tested this using three cases. The first and second case use different runs of the same ESM, except the physical circulations differ between them in the first case while the biological equations differ between them in the second. Our results indicate that the NNEs were capable of extracting the relationships between variables, allowing us to distinguish between differences due to changes in circulation (which do not change relationships) from changes in biogeochemical formulation (which do change relationships).

## 1 Introduction

Earth system models (ESMs) are increasingly used to help us understand how anthropogenic greenhouse gas emissions will affect biological systems and how such changes will feed back on the climate system. Although these methods provide an avenue for examining processes on a global scale, their representations of biological and physical processes of the natural world are limited by imperfect knowledge. As a result, estimates of critical biological and physical parameters can vary quite substantially. For example, from tracer experiments in the North Atlantic subtropical gyre, diapycnal diffusivity was estimated between 0.1 to 0.5 cm$^2$ s$^{-1}$ (Ledwell et al., 1998), with similar values having been used in ESMs. Varying the diapycnal diffusivity within this range in ESMs has been shown to yield different results in the biogeochemical output (Oschlies, 2001; Duteil and Oschlies, 2011). Furthermore, ESMs do not agree about how to represent phytoplankton growth parameters, such as temperature dependence. In the nine ESMs compared in





Laufkötter et al. (2015), the $Q_{10}$ value describing the sensitivity of growth rate to 10 degree increases in temperature ranged from 1.68 to 3, with some models varying the $Q_{10}$ values based on the size or type of phytoplankton.


The uncertainty associated with some ESM parameters can make it difficult to understand why different ESMs may yield different predictions for biological variables ranging from productivity to carbon uptake. Bopp et al. (2013) demonstrated that while CMIP5 models showed the same overall global trends under climate change for variables such as pH, sea surface temperature, $O_2$, and primary productivity, there were significant cross-model differences in $O_2$ and primary productivity on regional scales. Traditional methods used to estimate the sensitivity of ESMs often compare the spatial distributions of biological and physical variables from different runs of a single ESM to each other or to observations. However, occasionally changes in one parameter improve the simulation of one variable while degrading the simulation of another (see for example, Bahl et al. (2019), their Table 2). Other times errors in one variable are due to errors in another (i.e., getting a physical front in the wrong place may mean that the biomass has the wrong distribution). The intent of ESMs is to get the correct spatial distribution because the correct relationships between variables are being modelled. However, it's difficult to know if the correct relationships *are* being modelled. Thus, a method is needed in which we can evaluate whether different ESMs yield different projections because of fundamental differences in biogeochemical formulation or whether such differences are primarily due to differences in physical circulations and climate sensitivities.


It was previously demonstrated that neural network ensembles (NNEs) were able to extract relationships between biological forcings and outputs within a simplified biogeochemical model (Holder and Gnanadesikan, 2021a). NNEs were able to outperform other machine learning algorithms, such as random forests. More importantly, NNEs also had the benefits of being able to extrapolate outside the range of the training dataset and to provide a measure of their uncertainty in their predictions. Holder and Gnanadesikan (2021a) defined two types of relationships between environmental forcings and biological responses: intrinsic and apparent. *Intrinsic* relationships are those where the effect of one predictor variable on an outcome (target variable) can be examined, while maintaining other predictors at a constant level. An example of this would be examining how phytoplankton react to different nutrient concentrations in a nutrient growth experiment, while all other factors remain constant. For ESMs, an example might be the Michaelis-Menten relationships that represent how phytoplankton interact with nutrients. *Apparent* relationships are determined by how the intrinsic relationships interact across space and time, where individual variables are not controlled but may systematically co-vary. An example of this would be comparing satellite observations of phytoplankton distributions against monthly distributions of nutrients; where low phytoplankton concentrations may result both from low nutrients and high light in the summer in some locations, but also high nutrients and low light in the winter in other locations. As a result, the apparent relationships between nutrients and biomass would not resemble the intrinsic Michaelis-Menten curves coded in the ESM. A proof-of-concept application of NNEs coupled with sensitivity analyses at the end of Holder and Gnanadesikan (2021a) demonstrated the ability of NNEs to draw out the colimitations in a non-linear biogeochemical model and illustrated how these colimitations differed from the Michaelis-Menten curves programmed into the model.




The objective of this paper is to investigate whether the application of NNEs and sensitivity analyses can provide useful information for determining differences in ESM outputs. In general, there are three primary drivers that lead to differences in the output of ESMs: physical forcings, phytoplankton physiology, or combinations of these two. Before applying this method to outputs of multiple ESMs, we chose to investigate whether the method worked well on

different runs of a single ESM in which physical parameters were changed to produce different circulations. It was uncertain whether the NNEs would be able to pick out the same apparent relationships of the same ESM when there were differences between runs in the physical forcings and intrinsic biological equations (phytoplankton physiology). If different versions of an ESM, which have different circulations, still yield the same apparent relationships between light/nutrients and biomass, it would suggest that circulation changes do not produce new patterns of co-limitation.

Furthermore, it would suggest that differences in the apparent relationships of *different* ESMs could be partitioned between those due to different physical circulations and those with different representations of biology. For example, if one uses the apparent relationships from model A to predict the biomass from model B given the environmental parameters from model B, any differences should be due to differences in the biological formulation.

To investigate the extent to which NNEs could characterize differences across ESMs, we explored three cases:

1. We examined an ESM in which biomass was by construction a function of nutrients and light. Using three different runs of this ESM, we maintained identical intrinsic biological relationships, but varied the physical parameters controlling the circulation across the different runs. The objective of the first case was to quantify

the extent to which differences in physical circulation might affect the apparent relationships between predictor (light, nutrient and temperature) and target (biomass) variables found by NNEs. If models with different circulations produced differences in the apparent relationships, this would indicate that differences in circulation could push the biology into fundamentally new states. However, if the NNEs found the same apparent relationships between runs when the physical circulation was changing, this would indicate that the

primary effect of changing the circulation was simply to change the times and locations where different combinations of light and nutrients were found, rather than creating fundamentally new states.

2. We used the same ESM as that of Case 1, except we maintained similar physical circulations between runs and changed the intrinsic biological relationships. The objective of the second case was to quantify the ability of NNEs to detect differences in the apparent relationships when the intrinsic biological relationships between

model runs were different and to document the size of those differences.

3. For the final case, we looked at two *different* ESMs that had different biogeochemical codes but were run within the same physical model giving them identical physical circulations. The first ESM followed the framework of the ESMs in Cases 1 and 2, where biomass was a function of nutrients. The second ESM allowed biomass to be advected and diffused, making biomass a function of nutrients, light *and* physical circulation. The objective

of the third case was to apply the principles from Cases 1 and 2 to more standard ESMs, to quantify the extent

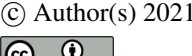



to which physical circulation contributes to these apparent relationships, and to identify challenges in comparing the apparent relationships between ESMs.

## 2 Methods

### 2.1 Earth System Models – Biogeochemical Codes

In general, biogeochemical components of ESMs predict the evolution of phytoplankton biomass using equations that have the general form

$$\frac{\partial B}{\partial t} + \vec{u} * \nabla B = \mu(N, Light, T) * B - G(B, \dots) + \nabla * \vec{K} * \nabla B \tag{1}$$

where $\vec{u}$ is the three-dimensional velocity field, $\mu$ is the phytoplankton growth rate which is a function of nutrients $N$, light and temperature $T$, $G(B, \dots)$ represents the grazing loss rate, which may be a function of phytoplankton biomass and/or other variables such as temperature or zooplankton concentration, and $\vec{K}$ is the three-dimensional mixing tensor.

Changes in physical parameters (for example changing the values in $\vec{K}$) would produce changes in transport of biomass. But the associated changes in circulation would also produce changes in other fields, such as $N$, $Light$, and $T$ (and thus in growth rate $\mu$). Differences in the physical parameters between models will produce both direct differences due to transport and indirect differences due to changes in growth and/or grazing. Additionally, insofar as the biology affects the absorption of shortwave radiation, it can produce differences in the circulation, although for
the simulations in this paper the differences are relatively small.

For this paper, we chose to focus on biogeochemistry components (BCs) run within two ESMs: Biogeochemistry with Light, Iron, Nutrients, and Gases (BLING) and Tracers of Phytoplankton with Allometric Zooplankton (TOPAZ). In general terms, BLING is a simplified version of TOPAZ. For Cases 1 and 2, we chose to only use model runs within
different versions of GFDL ESM2Mc, in which BLING is the BC with the reasoning that if the NNEs were unable to distinguish apparent relationships in the simpler BLING model, they would not be able to do so in the more complex TOPAZ model. In Case 3, we use versions of the GFDL ESM2M model in which BLING and TOPAZ are used as the BCs to compare apparent relationships found within the ESM.

### 2.2 Biogeochemistry with Light, Iron, Nutrients, and Gases (BLING)

BLING is a diagnostic biogeochemical model (Fig. 1) described in Galbraith et al. (2010), which was developed as a relatively cheap biogeochemical code that could be run in high-resolution models. Only four explicit tracers are included in within the model: oxygen, dissolved organic phosphorus, phosphate, and iron (the last two corresponding to the nutrients (N) in Fig. 1). Phytoplankton are represented as two size classes: small and large (Biomass (B) in Fig. 1). Phytoplankton growth and grazing $G(B, T)$ are modelled using the phytoplankton size-dependent loss equation
developed by Dunne et al. (2005) represented as

$$\mu(N, Light, T) * B \approx G(B, T) = \lambda \left(\frac{B}{P_*}\right)^\alpha B \tag{2}$$



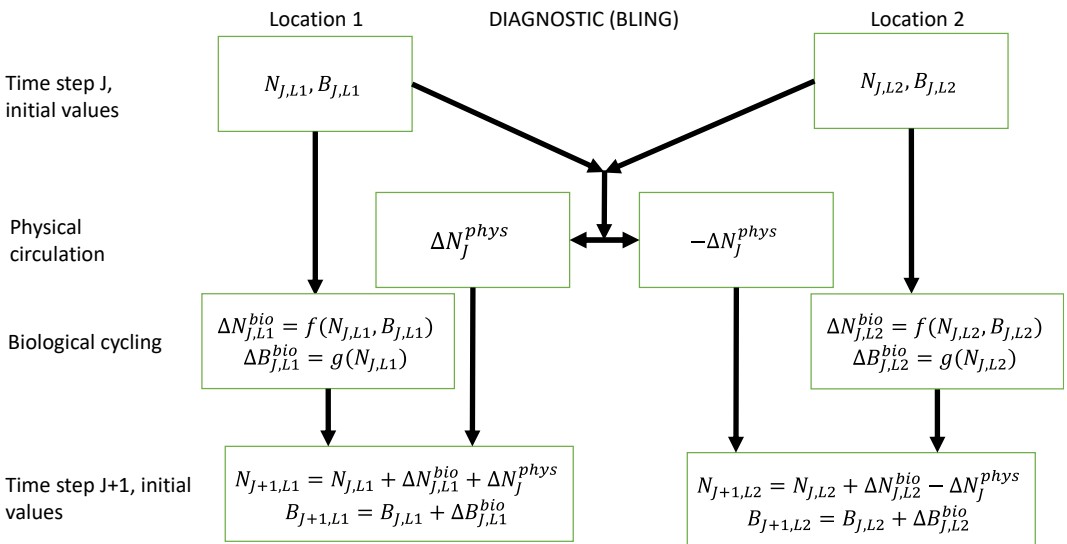

**Figure 1:** Conceptual diagram of how biogeochemical evolution is computed within an ESM using the BLING BC. The letters and abbreviations represent: nutrients (N), phytoplankton biomass (B), the physical circulation component (phys), and the biological cycling component (bio). Each location has initial values for nutrients and biomass. These initial values are passed to the intrinsic biological relationships which then feed into the *g* function in the biological cycling box that are then used to calculate the changes in nutrients and biomass due to biological cycling. The initial nutrient concentrations between the two locations result in a change in nutrients from physical transport, which is equal in magnitude and opposite in sign between the two boxes (physical circulation component). When the physical circulation and biological cycling portions are coupled together, the nutrients and biomass for the next time step are calculated.

where $\lambda$ is a grazing rate, $P_*$ is a biomass scaling for grazing, and $\alpha$ is a grazing exponent. The grazing rate includes all losses due to grazing, viral lysis, temperature-dependent loss, and others. For the small phytoplankton size class $\alpha$ = 1 and for the large phytoplankton size class $\alpha$= 1/3. This means the large phytoplankton biomass is more sensitive to environmental conditions that then small phytoplankton biomass. The growth rate ($\mu$) in Eq. (2) goes as

$$\mu = \mu_o \cdot \exp(kT) \cdot \min\left(\frac{Fe}{K_{Fe} + Fe}, \frac{PO_4}{K_{PO_4} + PO_4}\right) \cdot \left(1 - \exp\left(-\frac{Irr}{K_{Irr}}\right)\right) \tag{3}$$

where $\mu$ is the growth rate, $T$ is the temperature with constant $k = 0.063°\text{C}^{-1}$ following Eppley (1972), $K_{Fe,PO_4,Irr}$ are the half-saturation constants, and $Fe$, $PO_4$, and $Irr$ are the concentrations of dissolved iron, phosphate, and irradiance, respectively. $K_{Irr}$ is a function of the nutrient an temperature dependent growth rate. The time averaged biomass then goes as



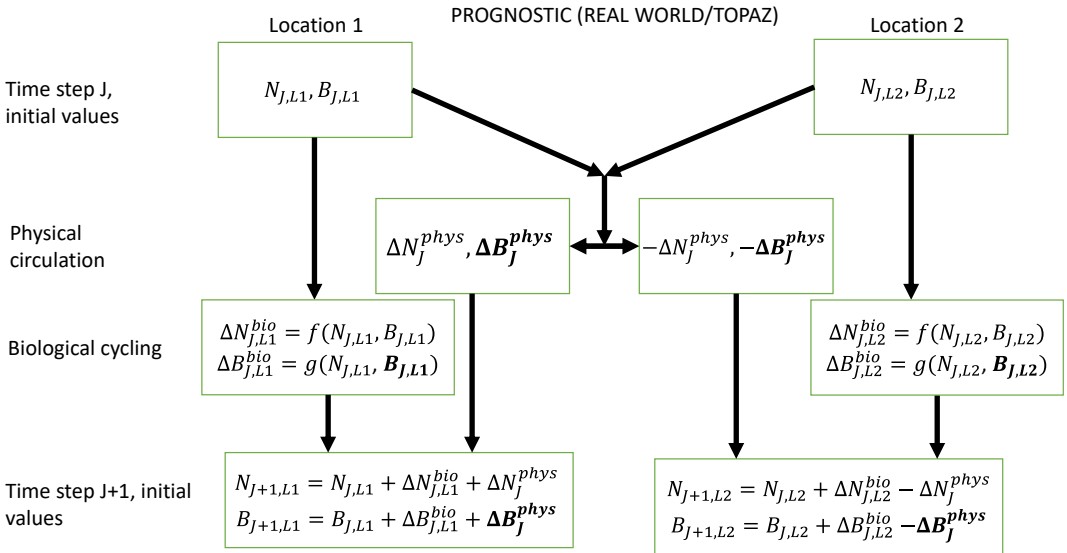

**Figure 2:** Conceptual diagram of how biogeochemical evolution is computed within an ESM using the prognostic TOPAZ BC. The letters and abbreviations represent: nutrients (N), phytoplankton biomass (B), the physical circulation component (phys), and the biological cycling component (bio). This ESM differs from the one described in Fig. 1. In this prognostic model, the changes in biomass from the biological cycling component are a function of the nutrients and biomass, rather than nutrients alone. Additionally, a change in biomass due to physical circulation is added.

$$\overline{B} \approx \left(\frac{\overline{\mu}}{\lambda}\right)^{\frac{1}{\alpha}} P_* \tag{4}$$

Note that this means that given *N*, *Light*, and *T* (all of which are still predicted by the circulation model), the apparent
relationships between biomass, nutrients, and light are potentially tightly coupled to the intrinsic relationships governing phytoplankton physiology that determine the growth rate.

### 2.3 Tracers of Phytoplankton with Allometric Zooplankton (TOPAZ)

TOPAZ is a prognostic biogeochemical model included in the Geophysical Fluid Dynamics Laboratory (GFDL) ESM2M (Dunne et al., 2013; Fig. 2). It includes a total of 30 tracers to model cycles such as nitrogen, phosphorus,
iron, oxygen, carbon, and others (Nutrients (N) in Fig. 2). TOPAZ models three phytoplankton groups (small, large, and diazotrophic; Biomass (B) in Fig. 2) with light limitation based on the equations of Geider et al. (1997). Additionally, phytoplankton loss/grazing and particle export are modelled using the same size-dependent formulation as those used in Eq. (2), though without imposing the quasi-equilibrium assumption that leads to Eq. (4). TOPAZ differs from BLING in its number of tracers (and associated limitations) and the allowance for advection/diffusion of



nutrients and biomass ($\Delta B_j^{phys}$ in Fig. 2). This means that the loss rate of phytoplankton in TOPAZ is effectively a

function of circulation as well the temperature and biomass-dependent grazing rate, $\lambda \left(\frac{B}{B_*}\right)^{\alpha}$. This will produce

different biomasses in locations that have the same growth rates. Additionally, a key difference between BLING and

TOPAZ is that the latter includes denitrification and nitrogen fixation. This then means (as suggested by Tyrrell

(1999)) that the nitrogen is the proximate limiting nutrient, while phosphorus is the ultimate limiting nutrient; a

distinction that is not made in BLING.

### 3 Case Descriptions

#### 3.1 Case 1 - Same ESM: Identical Biological Equations, Different Physical Circulations

The aim of Case 1 was to quantify the extent to which differences in physical circulations between model runs of the

same ESM with identical intrinsic biological relationships would affect the apparent relationships found by NNEs. As

stated in Section 2.1, we chose to compare versions of GFDL ESM2Mc in which BLING is configured identically so

we can be certain the differences are solely due to circulation changing the environmental conditions, and not the

phytoplankton loss rates.

We chose to use three configurations of GFDL ESM2Mc. The three model runs consisted of: a standard historical pre-

industrial model spin-up (BLING – PI Control), a similar case to the first but where the carbon dioxide concentration

was four times higher (BLING – 4x $CO_2$), and a case similar to the historical spin-up except that the horizontal mixing

parameter was three times higher (BLING – 3x Mixing). These model runs are described in greater detail in

Gnanadesikan et al. (2013), Pradal and Gnanadesikan (2014), and Bahl et al. (2020). With the standard historical

model essentially serving as a form of a "control," the two remaining cases allowed us to examine if changes in the

physical circulation would result in changes to the apparent relationships.

The predictor variables for each model run were macronutrient (ex. phosphate), micronutrient (ex. dissolved iron),

irradiance, and temperature. The target variables were small phytoplankton biomass and large phytoplankton biomass.

One NNE was trained for each target variable of each model run for a total of six NNEs in Case 1 (three model runs

and two target variables in each run). Details of the NNE training can be found in Section 2.3.

#### 3.2 Case 2 - Same ESM: Different Diagnostic Biological Equations, Near-Identical Physical Circulations

The purpose of Case 2 was to quantify the differences found by NNEs between the apparent relationships of model

runs from the same ESM when the biological equations differ between runs, but the physical circulations are nearly

identical.


As in Case 1, we again chose to use different configurations of ESM2Mc, but this time we keep the physical

parameterizations constant but change constants within the BLING BC. We used two model runs: the standard





historical pre-industrial model spin-up used in Case 1 (BLING – PI Control) and one with similar distributions to PI Control but different half-saturation coefficients ($K_{Fe}$ and $K_{PO4}$ in Eq. (3)) for small and large phytoplankton (BLING

– LgSm). Changing the half-saturation coefficients, which directly affects phytoplankton growth, is analogous to changing the biological equations. Relative to the PI Control, the half-saturation coefficients in LgSm were decreased by $\sqrt{3}$ for small phytoplankton and increased by $\sqrt{3}$ for large phytoplankton. While these changes produce small differences in circulation and SST ($R^2 = 0.9949$ for SST between the two model runs) via changing the absorption of shortwave radiation, these differences are small. The primary impact of these changes affects the distribution of

nutrients.

The predictor variables for the model runs of Case 2 were the same as those in Case 1 (macronutrient, micronutrient, irradiance, and temperature). Likewise, the target variables were also the same as those in Case 1 (small and large phytoplankton biomass). A total of four NNEs were trained for Case 2 (two model runs and two target variables).

**3.3 Case 3 - Different ESMs: Prognostic vs. Diagnostic Biological Equations, Identical Physical Circulations**

For Case 3, the goal was to examine whether the results from a diagnostic BC from Cases 1 and 2 still held when a prognostic BC was used. Our goal was to examine any differences in apparent relationships, along with identifying challenges when comparing apparent relationships across more realistic ESMs. In this experiment, the BCs were governed by different biological equations, but were run within the same physical model so that the temperatures and

light seen by the two BC codes were identical.

One of our model simulations uses a version of BLING as the BC, while the other uses TOPAZ. For the BLING model run, the iron concentrations were fixed at their climatological values since this formulation was previously used to develop a model for very high-resolution studies (miniBLING). We chose this pair of simulations as the miniBLING

code was run in an identical physical circulation to the TOPAZ model run and so the light and temperature experienced by the two model ecosystems are identical. As described in Galbraith et al. (2015), the output is from the ocean component of ESM2M forced with historical atmospheric forcing which we denote as ESM2Mo.

The predictor variables for Case 3 were limited to variables that were present in both ESMs: macronutrient,

micronutrient, and irradiance. The target variable was total biomass. The biomass was not split into small and large phytoplankton biomass because the miniBLING output only contained total biomass. For consistency, the small and large phytoplankton biomass values in TOPAZ were combined to give total biomass. Two NNEs were trained for Case 3 (two ESM runs and one target variable).

**3.4 Neural Network Ensembles (NNEs)**

Neural network ensembles (NNEs) are an ensemble machine learning (ML) method. NNEs are comprised of a collection of individual NNs where the predictions of each NN are averaged into a single prediction. This ensemble approach has been shown to outperform individual NNs and reduce the generalization error within a dataset (Hansen



and Salamon, 1990) by turning individual "weak learners" into a single "strong learner." Individual neural networks (NNs) can fit a non-linear function to a dataset without assuming any prior knowledge of the system. For a more thorough discussion of NNs, please refer to Schmidhuber (2015). The basic structure of the NN approach that we use here is described in Appendix 1 of Scardi (1996).

We chose to use NNEs for several reasons:

1.  The ensemble approach of NNEs allows us to view the uncertainty in any given prediction based on the individual predictions of each NN.
2.  NNEs possess some capability of extrapolating outside the range of the data on which they are trained.
3.  As recently shown in Holder and Gnanadesikan (2021a), NNEs were able to more closely reproduce the underlying intrinsic relationships compared to RFs, mainly because of their ability to extrapolate.

The structure of the individual NNs was consistent between the three cases with each NN containing 25 hidden nodes in the hidden layer with a hyperbolic tangent sigmoid activation function and 1 node in the output layer with a linear activation function. The only difference between each case was in the number of input nodes: Cases 1 and 2 each contained four input nodes (one for each predictor) and Case 3 contained three input nodes. For example, one NN of the NNE for the small phytoplankton biomass target variable in Case 1 would have the following structure:

1.  The four predictor variables for Case 1 (first column of Table 1) correspond to the four nodes in the input layer of the NN.
2.  Each of the four input nodes is connected by weights to each of the 25 nodes in the hidden layer. Additionally, a bias term is connected to each of the hidden nodes.
3.  Each of the nodes in the hidden layer is connected by weights to the single node in the output layer, which for this instance would correspond to the target variable of small phytoplankton biomass. As with the hidden layer, a bias term is connected to the single output node.

The training of each NN was carried out using the "feedforwardnet" function in MATLAB 2019b (MATLAB, 2019). The training was stopped when the error between the predictions and observations increased for six consecutive epochs.

Separate NNEs were trained for each response variable in each model run, which equated to six NNEs (2 target variables, 3 simulations) in Case 1, four NNEs in Case 2, and two NNEs in Case 3. For consistency, the same framework and settings were used for the construction of the NNEs with each one consisting of 25 individuals NNs.

It was demonstrated in Holder and Gnanadesikan (2021a) that the predictions produced by this approach were insensitive to the particular configuration of the NNEs. They tested various conditions that could affect the NNE performance including activation functions of the hidden layer nodes, number of hidden layers, and number of nodes





in the hidden layer. The settings specified here allowed for reasonable training times while maintaining high

performance metrics. For more detailed information, see Appendix B2 in Holder and Gnanadesikan (2021a).

Each variable was also scaled between -1 and 1 relative to each variable's maximum and minimum

$$V_S = \frac{max_S - min_S}{max_U - min_U} (V_U - min_U) + min_S \qquad (5)$$

Where V is the value of a variable being scaled, S (subscript) is the scaled value, and U (subscript) is the unscaled value. This scaling puts the predictor values in the same range, so more weight is not given to variables with larger

ranges. Additionally, this step decreases the training time of the NNs so that no values are too close to the limits of the hyperbolic tangent sigmoid activation function. The variables and predictions were then scaled back to their original values for analysis and presentation of the results (Eq. (6)). The letter representations in Eq. (6) are the same as those in Eq. (5).

$$V_U = \frac{max_U - min_U}{max_S - min_S} (V_S - min_S) + min_U \qquad (6)$$

When using ML, it is possible to produce overly complex relationships that "overfit" the data. This provides a good match for the data on which an ML model is trained but leads to poor predictions when new data is presented to the model. This can be avoided by splitting a dataset into training and testing subsets. For this manuscript, this means each NNE was trained using only the observations in the training subset and tested on the observations from the testing subset. The data from each model run was randomly split into training and testing subsets with 60% of the observations

from a dataset going to the training subset and the other 40% going to the testing subset. The observations set aside in the testing subset were ones that the NNEs never saw during their training phase. This provides a way to evaluate each trained NNE and its generalizability. If performance metrics of a trained NNE are similar between the training and testing subsets, it suggests that the variance of the dataset is well captured in the training phase and the NNE is generalizable to the entire dataset.


To assess the performance of each NNE, we calculated the $R^2$ values and root mean squared error (RMSE) by comparing the predictions from each NNE to the actual values within the respective training and testing subsets.

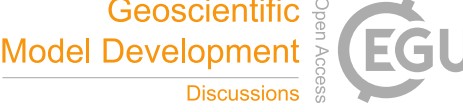

**Table 1:** Summary of each case which includes information on the predictor variables, the target variables, the ESMs, the model runs, the biological specifications, and the physical circulation specifications.

| Case # | Predictor Variables | Target Variables | Biogeochemical Component | Model Runs | Biological Specifications | Physics/Circulation Specifications |
|---|---|---|---|---|---|---|
| 1 | Macronutrient (mol kg$^{-1}$); Micronutrient (mol kg$^{-1}$); Irradiance (W m$^{-2}$); Temperature (°C) | Small Phytoplankton Biomass (mol kg$^{-1}$); Large Phytoplankton Biomass (mol kg$^{-1}$) | BLING | PI Control; 4xCO2; 3x Mixing | Identical diagnostic BC across model runs | Predicted by different versions of ESM2Mc produced by significant changes in phyical parameters |
| 2 | Macronutrient (mol kg$^{-1}$); Micronutrient (mol kg$^{-1}$); Irradiance (W m$^{-2}$); Temperature (°C) | Small Phytoplankton Biomass (mol kg$^{-1}$); Large Phytoplankton Biomass (mol kg$^{-1}$) | BLING | PI Control; LgSm | Different diagnostic BC across model runs | Nearly identical circulations produced by ESM2Mc |
| 3 | Macronutrient (mol kg$^{-1}$); Micronutrient (mol kg$^{-1}$); Irradiance (W m$^{-2}$) | Total Phytoplankton Biomass (mol kg$^{-1}$) | miniBLING and TOPAZ | One model run from miniBLING; one model run from TOPAZ | Simple diagnostic vs complex pronostic BC | Identical physical circulations produced by ocean component of ESM2M |

The NNEs in each case and matching size class were also asked to make predictions on the testing subsets of the other model runs. For example, in Case 1 the NNE trained on the small phytoplankton of PI Control was asked to make predictions for small phytoplankton of $4xCO_2$ using the values of the predictors from the testing subset of the $4xCO_2$ model run. These results were then compared to the actual values of the target variable to calculate the RMSE. This RMSE was then used to calculate the percent increase/decrease in error when compared against the RMSE calculated from a point-by-point comparison of each model run against the others. The purpose of this was to provide another metric for testing if the NNEs had found common apparent relationships across model runs. If an NNE trained on one model run was able to accurately predict the outcomes of the other model runs leading to a reduction in the RMSE, it would suggest that the NNE had found similar apparent relationships between the model runs. On the other hand, if it showed an increase in RMSE, it would suggest that the apparent relationships between the model runs were different in biologically important ways.

To view the apparent relationships found by the NNEs, we conducted sensitivity analyses in which we presented each NNE with a unique set of values for the predictors. Compared to spatiotemporal distributions and time series, sensitivity analyses allow for the visualization of relationships between predictor and target variables. In each sensitivity analysis, one predictor was varied across its minimum and maximum range, while the other variables were held at a specified value, such as each variable's 25th percentile. This was repeated for the 50th and 75th percentile values of each variable as well. This allowed us to visualize how the biomass predictions changed across one variable's range when the other variables were held at a specified value. An example of this would be varying the macronutrient concentration while holding the micronutrient, irradiance, and temperature variables at their 25th or 75th percentile





values. This allowed us to see how the macronutrient concentration affected biomass when other nutrients were low

or high, respectively.

## 4 Results and Discussion

### 4.1 Case 1 – Same ESM: Identical Biological Equations, Different Physical Circulations

In Case 1, our objective was to quantify the extent to which differences in physical circulation might affect the apparent relationships found by NNEs when the intrinsic biological relationships remained the same between the model runs

and the physical circulation parameters differed. It was uncertain whether changing the circulation would push the biology into fundamentally new states (i.e., different apparent relationships) or whether the physical circulation would simply act to change the location of where combinations of light and nutrients were found (ie. same apparent relationships).

Our results support the latter case, in that the locations of particular environments were simply being shuffled around. The sensitivity analysis showed that each NNE found similar apparent relationships between biomass and each of the predictors for the respective size classes, insofar as each line fell within the standard deviation of the others (Fig. 3 and 4). For example, the standard deviation (gray region) around the predicted apparent relationships for the large phytoplankton (dashed lines) all overlap one another (Fig. 3). The same can be seen for the predicted apparent

relationships for the small phytoplankton (Fig. 4). Additionally, we were confident in the apparent relationships since each NNE acquired high performance metrics in both the training and testing subsets (highest RMSE = $3.11 \times 10^{-9}$ mol $kg^{-1}$; Table 2) relative to the mean value of the total biomass ($1.24 \times 10^{-8}$ mol $kg^{-1}$).

This result can be better understood by considering the conceptual diagram of how the diagnostic BC BLING works

within an ESM (Fig. 1). For each time step, nutrients are calculated as a function of three terms: the initial nutrients, the change in nutrients from biology, and the change in nutrients from physical circulation. In contrast, the biomass is only a function of two terms: the initial biomass values and the change in biomass due to biological cycling. Thus, biomass is not directly affected by changes in the physical circulation. Additionally, this means that when given information on the biological predictors, but not the physical parameters, the NNEs were able to back out the apparent

relationships quite well.





**Table 2:** The performance metrics for the training and testing subsets for the trained NNEs from each case separated into their respective size classes and ESM/model runs.

| Case # | Phytoplankton Size | ESM/Model Run/BC | Training Data | | Testing Data | |
|---|---|---|---|---|---|---|
| | | | R-squared | RMSE | R-squared | RMSE |
| Case 1 | Small Phytoplankton | ESM2Mc / PI Control / BLING | 0.9912 | $6.24 \times 10^{-10}$ | 0.9908 | $6.35 \times 10^{-10}$ |
| | | ESM2Mc / 4x $CO_2$ / BLING | 0.9906 | $6.18 \times 10^{-10}$ | 0.9903 | $6.26 \times 10^{-10}$ |
| | | ESM2Mc / 3x Mixing / BLING | 0.9912 | $6.22 \times 10^{-10}$ | 0.9906 | $6.35 \times 10^{-10}$ |
| | Large Phytoplankton | ESM2Mc / PI Control / BLING | 0.9790 | $3.00 \times 10^{-9}$ | 0.9771 | $3.11 \times 10^{-9}$ |
| | | ESM2Mc / 4x $CO_2$ / BLING | 0.9749 | $2.74 \times 10^{-9}$ | 0.9740 | $2.77 \times 10^{-9}$ |
| | | ESM2Mc / 3x Mixing / BLING | 0.9804 | $3.00 \times 10^{-9}$ | 0.9778 | $3.11 \times 10^{-9}$ |
| Case 2 | Small Phytoplankton | ESM2Mc / PI Control / BLING | 0.9912 | $6.24 \times 10^{-10}$ | 0.9908 | $6.35 \times 10^{-10}$ |
| | | ESM2Mc / PI Control / BLING-LgSm | 0.9762 | $1.00 \times 10^{-9}$ | 0.9761 | $1.00 \times 10^{-9}$ |
| | Large Phytoplankton | ESM2Mc / PI Control / BLING | 0.9790 | $3.00 \times 10^{-9}$ | 0.9771 | $3.11 \times 10^{-9}$ |
| | | ESM2Mc / PI Control / BLING-LgSm | 0.9862 | $2.34 \times 10^{-9}$ | 0.9855 | $2.38 \times 10^{-9}$ |
| Case 3 | Total Phytoplankton | ESM2Mo / Historical / miniBLING | 0.9511 | $8.97 \times 10^{-9}$ | 0.9507 | $9.11 \times 10^{-9}$ |
| | | ESM2Mo / Historical / TOPAZ | 0.5893 | $8.97 \times 10^{-9}$ | 0.5867 | $8.99 \times 10^{-9}$ |

That similar apparent relationships were found between the model runs was further supported when we tasked each trained NNE with making predictions on the testing subsets of the other model runs for the same size class. For example, the NNE trained on the PI Control for small phytoplankton was tasked with making predictions for the small

phytoplankton biomass of 4xCO₂ and 3xMixing using the predictor values from their testing subsets. This test allowed for the evaluation of the robustness of the relationships that each NNE found. If the NNEs were finding different relationships between the model runs, the NNE from one model run would likely perform poorly when predicting on the other model runs. Our results show that the NNEs performed well when applied to the other model runs (highest RMSE = $3.74 \times 10^{-9}$ mol kg$^{-1}$; Table 3) relative to the average value of total biomass ($1.24 \times 10^{-8}$ mol kg$^{-1}$). Given that

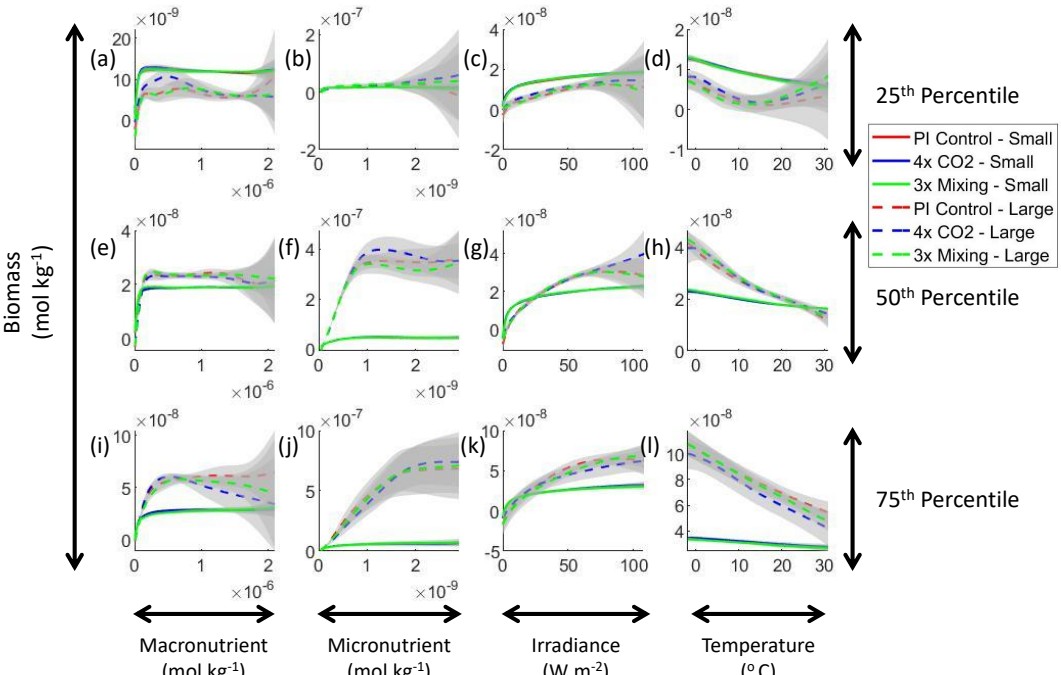

**Figure 3:** Sensitivity analysis plots for the small and large phytoplankton of Case 1. Each line is the prediction for the NNE specific to each model run and the color of each line represents the model run (PI Control – Red; $4xCO_2$ – Blue; 3xMixing - Green). The solid lines correspond to the small phytoplankton and the dashed lines to the large phytoplankton. The gray region around each line shows one standard deviation in the predictions of the individual NNs that make up each NNE (ex. The gray region around the solid red curves shows the standard deviation in the predictions of the 25 NNs that make up that particular NNE). The rows correspond to the percentile value at which the other predictor variables were held constant (ex. Box (a) varies the macronutrient across its min-max range and holds the micronutrient, irradiance, and temperature at their respective 25th percentile values). Columns show the x-axis variables as they vary between their min-max range. The y-axis in all subplots is the biomass concentration. Note that the biomass scale changes with each subplot.

these values are close to the performance metrics of their original datasets (Table 2 vs Table 3), it seems logical to say that this was because they were finding the same relationships.



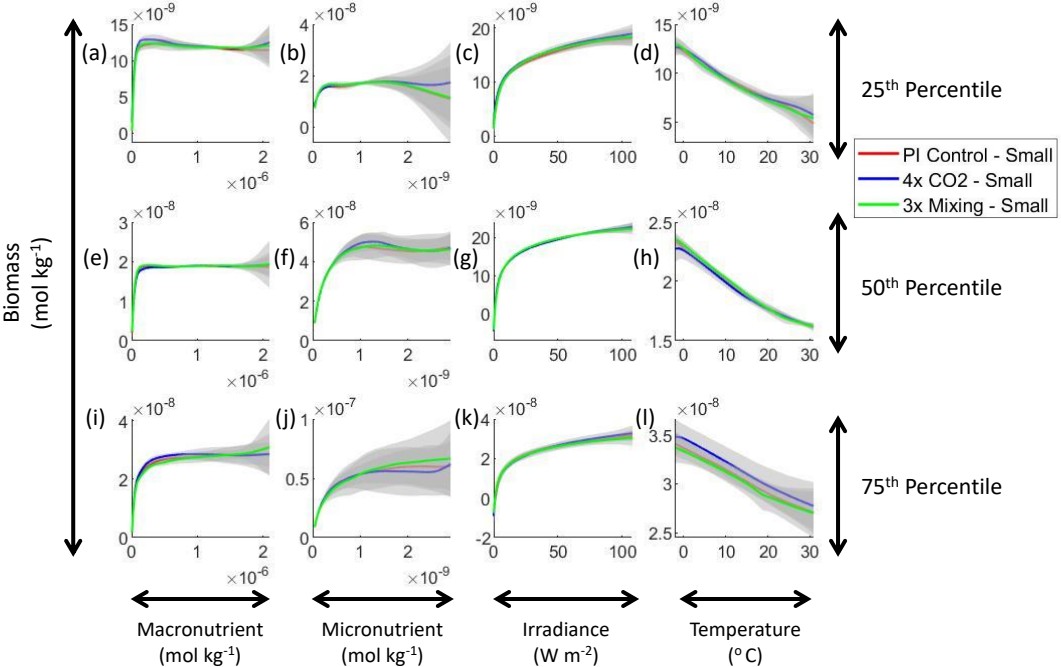

**Figure 4:** Sensitivity analysis plots for the small phytoplankton of Case 1. This figure is provided to allow for examination of the apparent relationships for the small phytoplankton, since the large phytoplankton apparent relationships made it difficult to see those for the small phytoplankton in Fig. 3. Each line is the prediction for the NNE specific to each model run and the color of each line represents the model run (PI Control – Red; 4xCO2 – Blue; 3xMixing - Green). The gray region around each line shows one standard deviation in the predictions of the individual NNs that make up each NNE (ex. The gray region around the solid red curves shows the standard deviation in the predictions of the 25 NNs that make up that particular NNE). The rows correspond to the percentile value at which the other predictor variables were held constant (ex. Box (a) varies the macronutrient across its min-max range and holds the micronutrient, irradiance, and temperature at their respective 25th percentile values). Columns show the x-axis variables as they vary between their min-max range. The y-axis in all subplots is the biomass concentration. Note that the biomass scale changes with each subplot.

Additionally, using the NNEs to predict the other runs led to decreases in error relative to the error from comparing each run against the others. For example, the initial point-by-point comparison of $4xCO_2$ and PI Control for small phytoplankton (Fig. 5 d) showed an RMSE of $3.06 \times 10^{-9}$ mol $kg^{-1}$, while using the NNEs from each model run to predict the other saw the RMSE go down with a reduction in error of about 76% (Table 3). This reduction of error was consistent across the other model runs and size classes with error reductions varying from 54-79% (Table 3). This






**Table 3:** The performance metrics for the NNEs being used to predict the outcome of the other model runs for the same size class of Case 1. In the top half of the table, the R-squared and RMSE are listed. The values in paratheses are the values from comparing the respective cases against one another (these are the same values listed in the respective scatter plots of Fig. 5 and 6). The values outside the parentheses are the values from using the trained NNE of the model listed in the row to predict the outcome of the model run in the column (ex. The NNE trained on $4xCO_2$ was used to predict the PI Control outcome using the predictor values of PI Control. These values were compared against the actual values of the PI Control to compute the RMSE of $7.15x10^{-10}$). In the bottom half of the table is the percent decrease in RMSE from the number listed inside the parentheses to the RMSE outside the parentheses.

| | | | | Case being predicted | | | | | |
| --- | --- | --- | --- | --- | --- | --- | --- | --- | --- |
| | | | | Small Phytoplankton | | | Large Phytoplankton | | |
| | | | | PI Control | 4x CO2 | 3x Mixing | PI Control | 4x CO2 | 3x Mixing |
| **R-squared** | NNE being used for predicting | Small Phytoplankton | PI Control | - | (0.829) 0.9874 | (0.9287) 0.9902 | - | - | - |
| | | | 4x CO2 | (0.829) 0.9887 | - | (0.788) 0.9878 | - | - | - |
| | | | 3x Mixing | (0.9287) 0.9901 | (0.788) 0.9849 | - | - | - | - |
| | | Large Phytoplankton | PI Control | - | - | - | - | (0.7842) 0.9683 | (0.8831) 0.9772 |
| | | | 4x CO2 | - | - | - | (0.7842) 0.9722 | - | (0.7306) 0.969 |
| | | | 3x Mixing | - | - | - | (0.8831) 0.9738 | (0.7306) 0.963 | - |
| **RMSE** | NNE being used for predicting | Small Phytoplankton | PI Control | - | $(3.06 \times 10^{-9})$ $7.38 \times 10^{-10}$ | $(1.84 \times 10^{-9})$ $6.55 \times 10^{-10}$ | - | - | - |
| | | | 4x CO2 | $(3.06 \times 10^{-9})$ $7.15 \times 10^{-10}$ | - | $(3.56 \times 10^{-9})$ $7.3 \times 10^{-10}$ | - | - | - |
| | | | 3x Mixing | $(1.84 \times 10^{-9})$ $6.64 \times 10^{-10}$ | $(3.56 \times 10^{-9})$ $7.97 \times 10^{-10}$ | - | - | - | - |
| | | Large Phytoplankton | PI Control | - | - | - | - | $(1 \times 10^{-8})$ $3.11 \times 10^{-9}$ | $(7.34 \times 10^{-9})$ $3.2 \times 10^{-9}$ |
| | | | 4x CO2 | - | - | - | $(1 \times 10^{-8})$ $3.44 \times 10^{-9}$ | - | $(1.17 \times 10^{-8})$ $3.74 \times 10^{-9}$ |
| | | | 3x Mixing | - | - | - | $(7.34 \times 10^{-9})$ $3.34 \times 10^{-9}$ | $(1.17 \times 10^{-8})$ $3.33 \times 10^{-9}$ | - |
| **Percent Decrease in Error** | NNE being used for predicting | Small Phytoplankton | PI Control | - | 75.90% | 64.45% | - | - | - |
| | | | 4x CO2 | 76.66% | - | 79.53% | - | - | - |
| | | | 3x Mixing | 63.98% | 77.64% | - | - | - | - |
| | | Large Phytoplankton | PI Control | - | - | - | - | 69.09% | 56.32% |
| | | | 4x CO2 | - | - | - | 65.71% | - | 67.99% |
| | | | 3x Mixing | - | - | - | 54.45% | 71.50% | - |

implies the NNEs applied to the other runs were better able to predict the outcome than the point-by-point analysis, once again reinforcing our previous result.



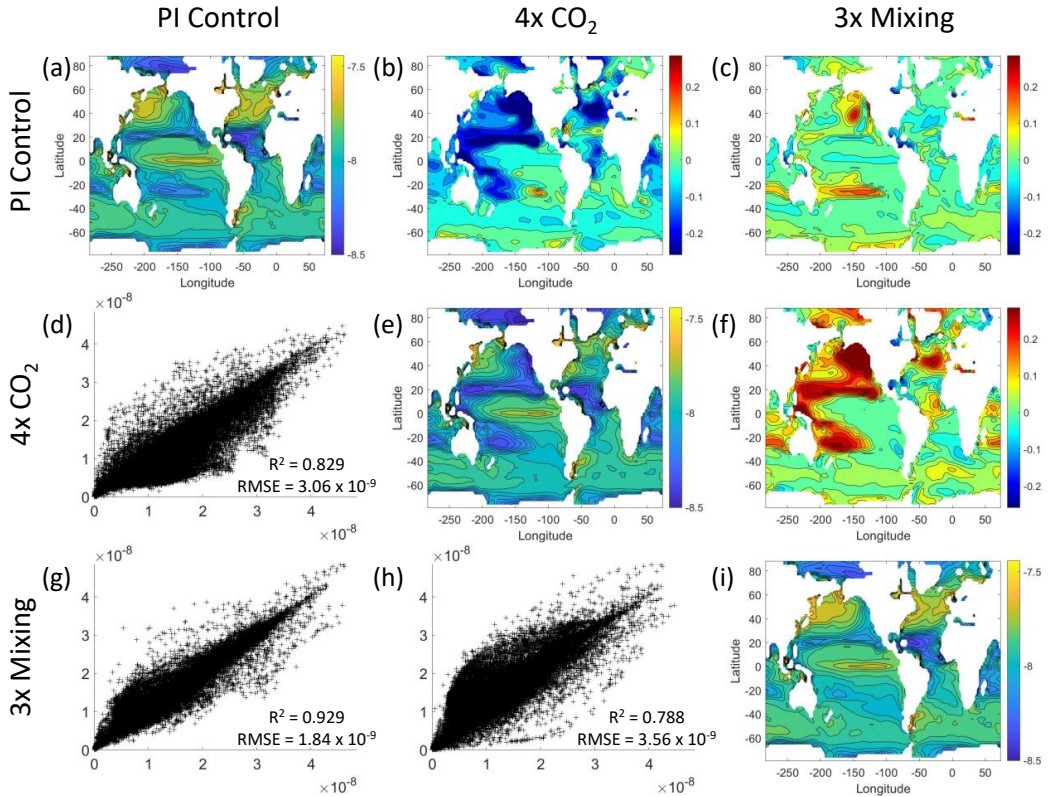

**Figure 5:** Comparison of the model runs for small phytoplankton biomass in Case 1. The units for biomass in all subplots are mol kg$^{-1}$. The subplots show point-by-point scatter plots comparing the model runs against one another (d, g, h), yearly averaged log10 biomass plots for each model run (a, e, i), and the log10 relative ratios comparing the yearly averaged contour plots of the model runs (b, c, f). The x-axis and y-axis of the scatter plots (d, g, h) correspond to the horizonal/vertical model run labels, respectively (ex. Box (d) shows PI Control on the x-axis and $4xCO_2$ on the y-axis). The yearly averaged log10 contour plots (a, e, i) correspond to the matching horizontal/vertical model run labels (ex. Box (a) shows the yearly averaged log10 biomass of PI Control). The log10 relative ratios (b, c, f) were calculated as the model run listed on the horizontal axis divided by the model run listed on the vertical axis (ex. Box (b) shows $4xCO_2$ divided PI Control).

That the NNEs from one model run were able to reproduce the results from the other model runs was not simply due to the models producing similar spatiotemporal patterns. To ensure that distinct differences between the model runs were present, we compared each model run against the others (Fig. 5 and 6). Differences in the biomass values between the three model runs were evident (Fig. 5 and 6). First, we compared each model run against the others in a point-by-point analysis and observed that different biomasses were occurring at the same spatiotemporal locations (Fig. 5 and

6 d, g, h). For example, in the small phytoplankton scatter plot for PI Control vs 4xCO2, PI Control showed a tendency



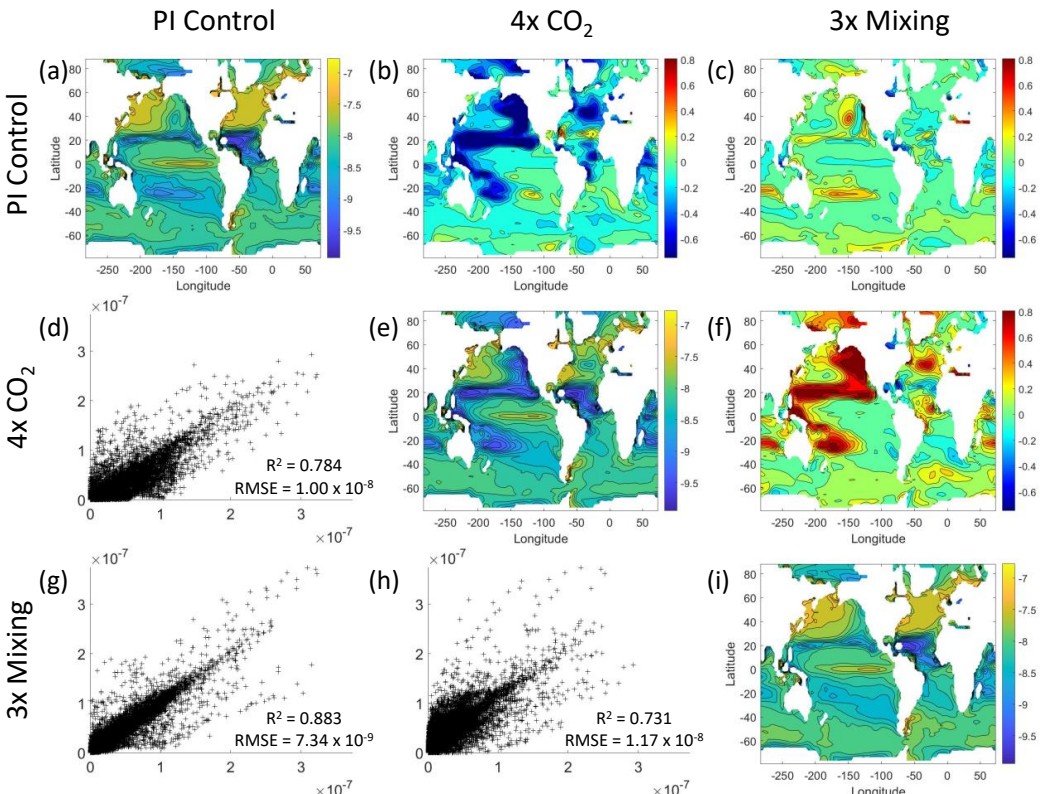

**Figure 6:** Comparison of the model runs for large phytoplankton biomass in Case 1. The units for biomass in all subplots are mol kg$^{-1}$. The subplots show point-by-point scatter plots comparing the model runs against one another (d, g, h), yearly averaged log10 biomass plots for each model run (a, e, i), and the log10 relative ratios comparing the yearly averaged contour plots of the model runs (b, c, f). The x-axis and y-axis of the scatter plots (d, g, h) correspond to the horizonal/vertical model run labels, respectively (ex. Box (d) shows PI Control on the x-axis and $4xCO_2$ on the y-axis). The yearly averaged log10 contour plots (a, e, i) correspond to the matching horizontal/vertical model run labels (ex. Box (a) shows the yearly averaged log10 biomass of PI Control). The log10 relative ratios (b, c, f) were calculated as the model run listed on the horizontal axis divided by the model run listed on the vertical axis (ex. Box (b) shows $4xCO_2$ divided PI Control).

of having higher biomass values than 4xCO2 across most locations (Fig. 5 d). Additionally, we looked at the contour plots and log relative ratios using the yearly averaged biomass for each case (Fig. 5 and 6 a-c, e, f, i). Specific large differences that we noted were higher biomass in the Pacific and Northern Atlantic in PI Control and 3xMixing relative to 4xCO$_2$ (Fig. 5 and 6 b, f) and the highest biomass in occurring in 3xMixing in the subtropical regions of the Pacific (Fig. 5 and 6 c). Similar patterns were observed in the large phytoplankton, as well (Fig. 6). These differences between






the model runs are relatively large (exceeding a factor of three in some locations) and allow us to dismiss the possibility that the similar apparent relationships were only due to strong similarities between the model runs.

Although the sensitivity analysis allowed us to see that the apparent relationships were the same for each size class, it
also allows us to see how the two size classes react differently to the same conditions. Most notably, the large phytoplankton seem to be very sensitive to the micronutrient compared to the small phytoplankton (Fig. 3; closer view of small phytoplankton in Fig. 4). When the other predictors are held at their 75th percentile values (high macronutrient, high irradiance, and warm temperature), the large phytoplankton are able to reach biomass values almost an order of magnitude higher than the small phytoplankton (Fig. 3 and 4 j). This is what would be expected given the cubic
relationship of large phytoplankton with growth rate. Another interesting relationship is the stark asymptotes in both size classes of the macronutrient plots, suggesting limitations by other nutrients, likely the micronutrient (Fig. 3 a, e, i). One unexpected relationship was the decreasing biomass with increasing temperature in both size classes (Fig. 3 d, h, l). This could be a result of warmer regions having less available nutrients or because of the temperature dependent Chl:C ratios which would lead to phytoplankton needing more light in warmer waters.


Relative to our main objective in Case 1 to quantify the extent to which differences in physical circulation affect the apparent relationships, our results indicated that the different physical circulations did not produce differences in the apparent relationships found by NNEs. When the biological equations remained the same, changing the physical parameters simply changed where combinations of nutrients and light occurred. In contrast to changes in nutrients,
changes in biomass in the BLING ESM were not a function of the physical circulation.

### 4.2 Case 2 – Same ESM: Different Diagnostic Biological Equations, Near-Identical Physical Circulations

In Case 1, it was clear from our results that when the biological cycling was identical between model runs, the NNEs found the same apparent relationships because the biomass was not a function of the physical circulation. Since the biomass is clearly a function of the biological equations, it would be reasonable to assume that the apparent
relationships would be different between model runs that are governed by different biological equations. So, for Case 2, the objective was to quantify the extent to which NNEs could detect differences in the apparent relationships when the intrinsic biological relationships between model runs were different, while maintaining similar physical circulations and still using a diagnostic model which guarantees that identical nutrient, light, and temperature at two different points will produce identical biomass.


Our results show that NNEs can differentiate the apparent relationships between model runs when the biological equations differ. The sensitivity analysis for Case 2 shows that different apparent relationships were found between model runs and within the same size classes, relative to the non-overlapping gray standard deviation regions around each line (Fig. 7 and 8). Additionally, we can be fairly confident in these predictions given the high-performance
metrics in both the training and testing subsets (highest RMSE = $3.11 \times 10^{-9}$ mol kg$^{-1}$ [Table 2] vs. the average total biomass of $1.36 \times 10^{-8}$ mol kg$^{-1}$).

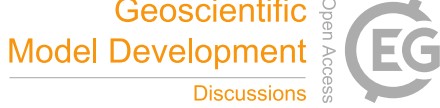

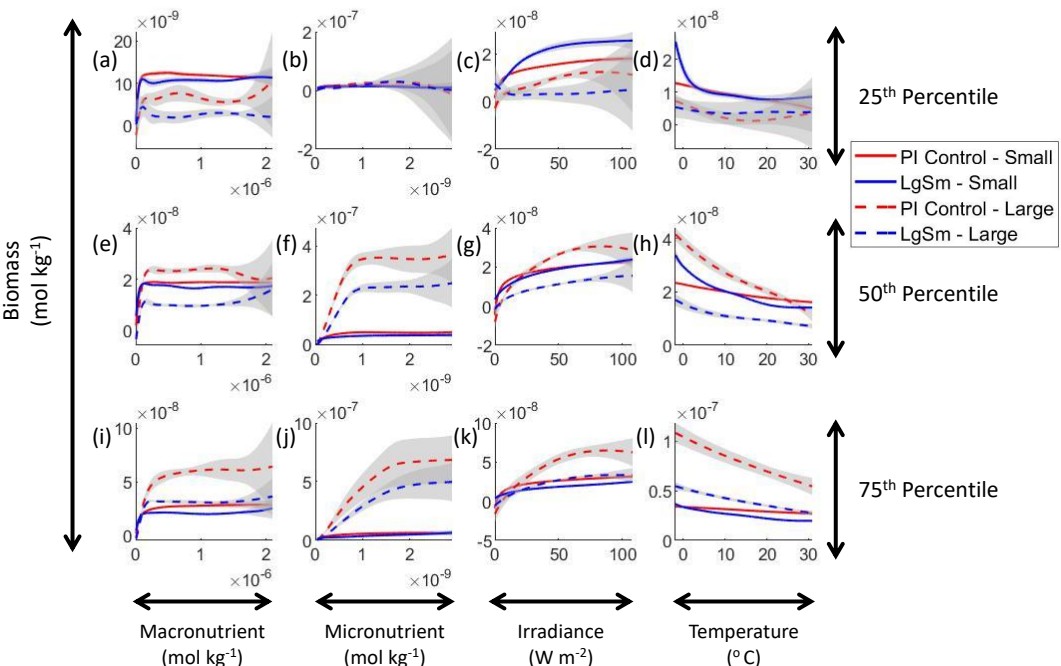

**Figure 7:** Sensitivity analysis plots for the small and large phytoplankton of Case 2. Each line is the prediction for the NNE specific to each model run and the color of each line represents the model run (PI Control – Red; LgSm – Blue). The solid lines correspond to the small phytoplankton and the dashed lines to the large phytoplankton. The gray region around each line shows one standard deviation in the predictions of the individual NNs that make up each NNE (ex. The gray region around the solid red curves shows the standard deviation in the predictions of the 25 NNs that make up that particular NNE). The rows correspond to the percentile value at which the other predictor variables were held constant (ex. Box (a) varies the macronutrient across its min-max range and holds the micronutrient, irradiance, and temperature at their respective 25[th] percentile values). Columns show the x-axis variables as they vary between their min-max range. The y-axis in all subplots is the biomass concentration. Note that the biomass scale changes with each subplot.

This result of different relationships, when the model runs are governed by different biological equations, reinforces what we found in Case 1. Changing the biological equations can be likened to changing how the nutrients affect the phytoplankton biomass (the function $g(N_{J,L1,L2})$ in Fig. 1). While it might seem obvious that changing the biological equations will change the biomass values, it remained unclear whether NNEs would be able to pick out these differences in the apparent relationships.




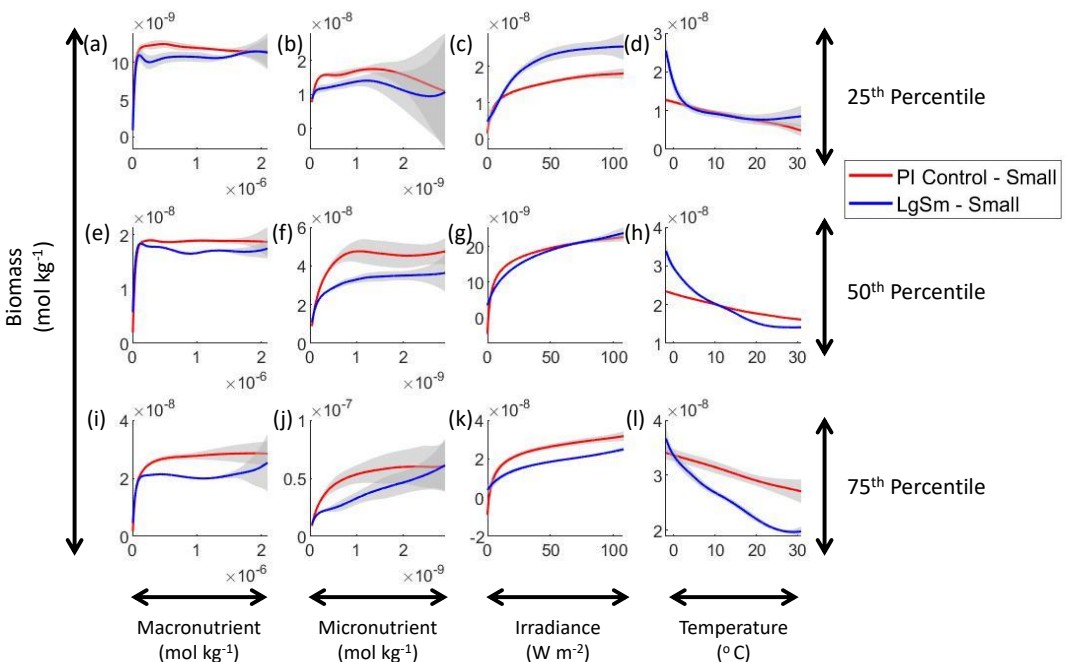

**Figure 8:** Sensitivity analysis plots for the small phytoplankton of Case 2. This figure is provided to allow for examination of the apparent relationships for the small phytoplankton, since the large phytoplankton apparent relationships made it difficult to see those for the small phytoplankton in Fig. 7. Each line is the prediction for the NNE specific to each model run and the color of each line represents the model run (PI Control – Red; LgSm – Blue). The gray region around each line shows one standard deviation in the predictions of the individual NNs that make up each NNE (ex. The gray region around the solid red curves shows the standard deviation in the predictions of the 25 NNs that make up that particular NNE). The rows correspond to the percentile value at which the other predictor variables were held constant (ex. Box (a) varies the macronutrient across its min-max range and holds the micronutrient, irradiance, and temperature at their respective 25th percentile values). Columns show the x-axis variables as they vary between their min-max range. The y-axis in all subplots is the biomass concentration. Note that the biomass scale changes with each subplot.

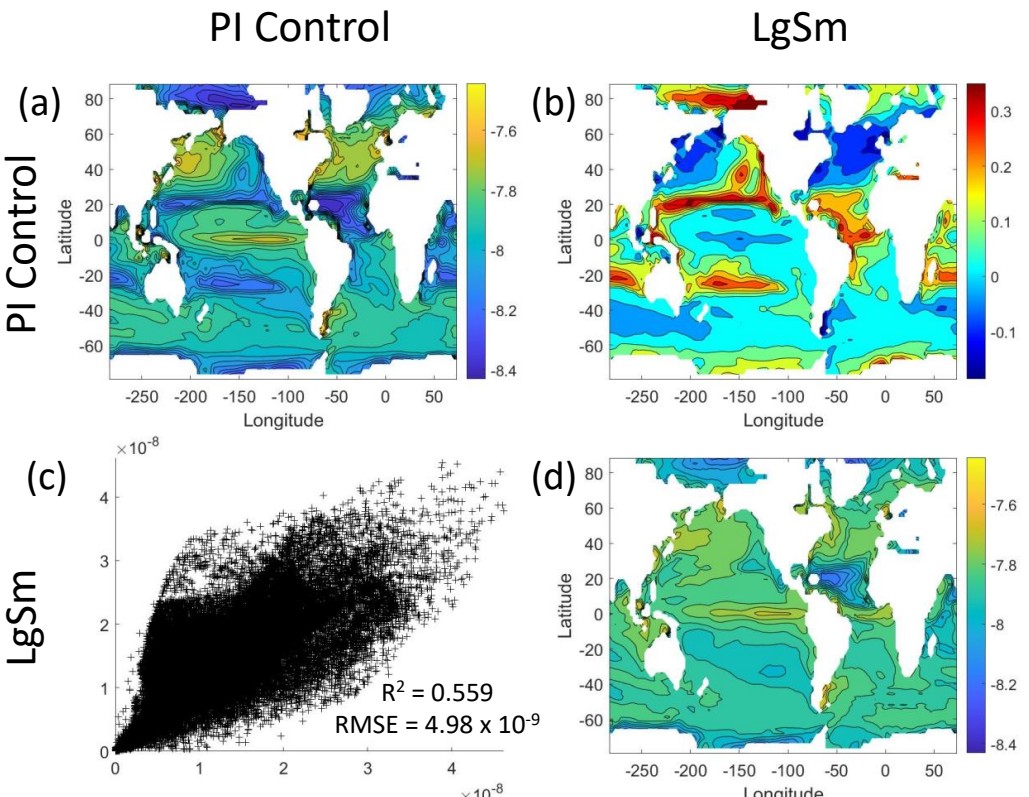

**Figure 9:** Comparison of the model runs for small phytoplankton biomass in Case 2. The units for biomass in all subplots are mol kg$^{-1}$. The subplots show point-by-point scatter plots comparing the model runs against one another (c), yearly averaged log10 biomass plots for each model run (a and d), and the log10 relative ratios comparing the yearly averaged contour plots of the model runs (b). The x-axis and y-axis of the scatter plots (c) correspond to the horizonal/vertical model run labels, respectively (ex. Box (c) shows PI Control on the x-axis and LgSm on the y-axis). The yearly averaged log10 contour plots (a and d) correspond to the matching horizontal/vertical model run labels (ex. Box (a) shows the yearly averaged log10 biomass of PI Control). The log10 relative ratios (b) were calculated as the model run listed on the horizontal axis divided by the model run listed on the vertical axis (ex. Box (b) shows LgSm divided PI Control).

We wanted to ensure there were noticeable differences between the model runs (Fig. 9 and 10). We did this in Case 1
to ensure that the similar apparent relationships found by the NNEs were not simply because of similarities in the model output. In Case 2, the difference in model outputs serves to reinforce the different apparent relationships found by the NNEs. In the point-by-point comparison, the large phytoplankton showed more agreement between model runs (Fig. 10 c) than the small phytoplankton (Fig. 9 c). However, when we examined the contour and log relative ratios

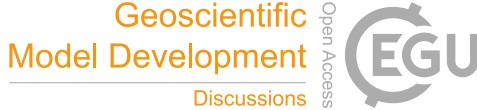

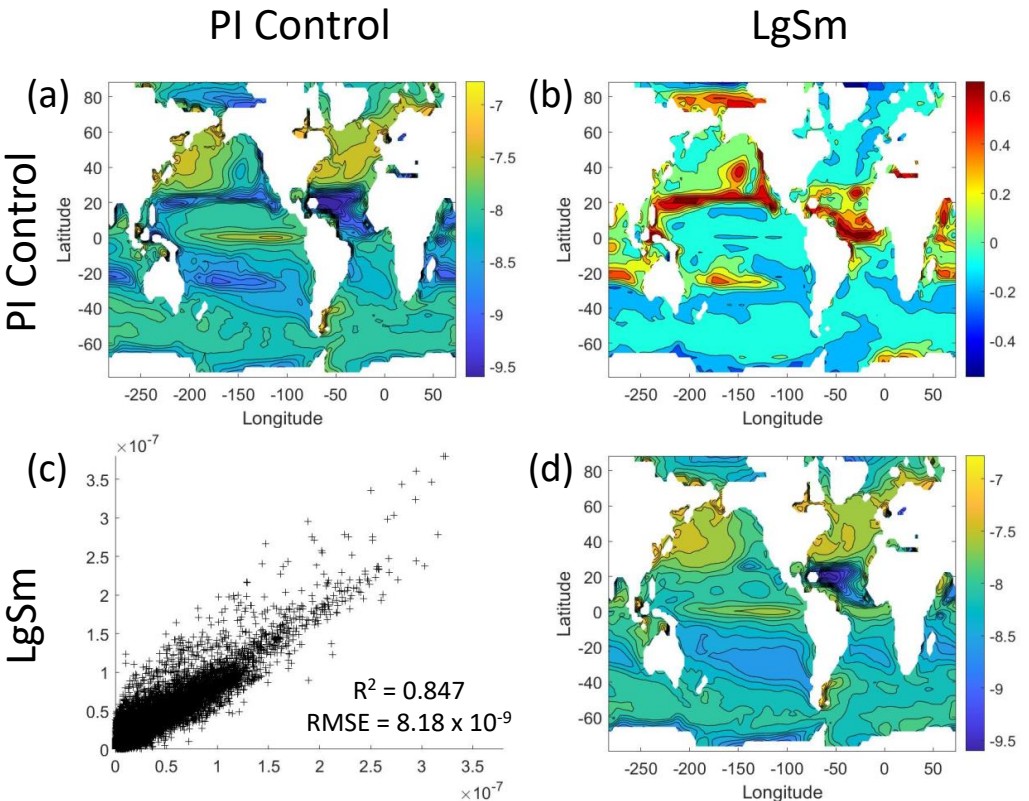

**Figure 10:** Comparison of the model runs for large phytoplankton biomass in Case 2. The units for biomass in all subplots are mol kg$^{-1}$. The subplots show point-by-point scatter plots comparing the model runs against one another (c), yearly averaged log10 biomass plots for each model run (a and d), and the log10 relative ratios comparing the yearly averaged contour plots of the model runs (b). The x-axis and y-axis of the scatter plots (c) correspond to the horizonal/vertical model run labels, respectively (ex. Box (c) shows PI Control on the x-axis and LgSm on the y-axis). The yearly averaged log10 contour plots (a and d) correspond to the matching horizontal/vertical model run labels (ex. Box (a) shows the yearly averaged log10 biomass of PI Control). The log10 relative ratios (b) were calculated as the model run listed on the horizontal axis divided by the model run listed on the vertical axis (ex. Box (b) shows LgSm divided PI Control).

(Fig. 9 and 10 a, b, d), it was evident that clear differences existed between the model runs. Both the small and large
phytoplankton showed higher concentrations in the LgSm model run compared to PI Control for the subtropical and
polar regions of the Pacific and Indian Oceans, along with higher concentrations in the equatorial Atlantic (Fig. 9 and
10 b).



Although the gray regions overlap toward the higher concentrations of each predictor, this is likely due to the lack of
observations in the training data meeting that criteria, without which the NNEs could not be constrained. For example,
in Fig. 7 (j), the apparent relationships of the large phytoplankton overlap past about 5 x $10^{-10}$ mol kg$^{-1}$ of the
micronutrient, because there were no observations in the training data that were greater than 5 x $10^{-10}$ mol kg$^{-1}$ of the
micronutrient while simultaneously being at the 75$^{th}$ percentile level of the macronutrient, irradiance, and temperature
(data not shown). Without observations to constrain them, the NNEs were unable to be constrained and, therefore, less
certain about the relationships in those regions which lead to higher uncertainty and overlapping standard deviations.
As in Case 1, our result was supported by the additional test in which the NNEs trained on one model run were tasked
with making predictions on the other. Had the NNEs found similar apparent relationships, the reductions in error
would have been of similar magnitude as those in Case 1 (Table 3 vs Table 4). For this second case, we saw that there
were only modest decreases in RMSE for the small phytoplankton and increases in RMSE for large phytoplankton
(Table 4). For example, relative to the RMSE of the point-by-point comparison, the RMSE decreased about 21% when
LgSm made predictions on PIControl for the small phytoplankton (Table 4). Additionally, it was observed that even
though the RMSE increased in the large phytoplankton, the $R^2$ values improved in the cross-model comparison
compared to the point-by-point comparison (0.92-0.93 vs 0.85; Table 4). This suggests that the NNEs improved the
simulation in terms of the overall pattern, but not the magnitude. These results make sense since the apparent
relationships of the small phytoplankton showed greater similarities than the apparent relationships of the large
phytoplankton (Fig. 7).

With respect to the apparent relationships that the NNEs uncovered, the large phytoplankton once again appeared to
be more sensitive to the micronutrient concentrations compared to the small phytoplankton (Fig. 7 b, f, j). Both size
classes showed asymptotes around the same concentrations for the macronutrient, albeit at different biomass values
(Fig. 7 a, e, i). As with Case 1, the decreasing biomass with increasing temperature was an unexpected relationship
(Fig. 7 d, h, l), which might be explained by the temperature dependent Chl:C ratios causing phytoplankton in warmer
regions to need more light.

Our main objective with Case 2 was to quantify the extent to which NNEs could detect differences in the apparent
relationships when the physical conditions between model runs were identical and the biological relationships differed.
With the biomass being a function of changes in biomass from biology (ie. the equations governing how nutrients
affect biomass), different biological equations produced differences in biomass. What was unclear was whether NNEs
would be able to highlight these differences in the apparent relationships. Our results indicate that NNEs could find
noticeable differences in the apparent relationships, insofar as the standard deviation regions did not often overlap in
the sensitivity analysis curves.



**Table 4:** The performance metrics for the NNEs being used to predict the outcome of the other model runs for the same size class of Case 2. In the top half of the table, the R-squared and RMSE are listed. The values in paratheses are the values from comparing the respective cases against one another (these are the same values listed in the respective scatter plots of Fig. 9 and 10). The values outside the parentheses are the values from using the trained NNE of the model listed in the row to predict the outcome of the model run in the column (ex. The NNE trained on LgSm was used to predict the PI Control outcome using the predictor values of PI Control. These values were compared against the actual values of the PI Control to compute the RMSE of $3.07 \times 10^{-9}$). In the bottom half of the table is the percent decrease in RMSE from the number listed inside the parentheses to the RMSE outside the parentheses (a negative percent means that the error increased).

| | | | | Case being predicted | | | |
| --- | --- | --- | --- | --- | --- | --- | --- |
| | | | | Small Phytoplankton | | Large Phytoplankton | |
| | | | | PI Control | LgSm | PI Control | LgSm |
| **R-squared** | NNE being used for predicting | Small Phytoplankton | PI Control | - | (0.5591) 0.8192 | - | - |
| | | | LgSm | (0.5591) 0.7899 | - | - | - |
| | | Large Phytoplankton | PI Control | - | - | - | (0.8465) 0.9334 |
| | | | LgSm | - | - | (0.8465) 0.9171 | - |
| **RMSE** | NNE being used for predicting | Small Phytoplankton | PI Control | - | $(4.98 \times 10^{-9})$ $3.95 \times 10^{-9}$ | - | - |
| | | | LgSm | $(4.98 \times 10^{-9})$ $3.07 \times 10^{-9}$ | - | - | - |
| | | Large Phytoplankton | PI Control | - | - | - | $(8.18 \times 10^{-9})$ $1.56 \times 10^{-8}$ |
| | | | LgSm | - | - | $(8.18 \times 10^{-9})$ $1.01 \times 10^{-8}$ | - |
| **Percent Decrease in Error** | NNE being used for predicting | Small Phytoplankton | PI Control | - | 20.59% | - | - |
| | | | LgSm | 38.20% | - | - | - |
| | | Large Phytoplankton | PI Control | - | - | - | -90.87% |
| | | | LgSm | - | - | -23.11% | - |

## 4.3 Case 3 – Different ESMs: Prognostic vs. Diagnostic Biological Equations, Identical Physical Circulations

From Cases 1 and 2, we learned from our results that NNEs were capable of discerning differences in apparent relationships between model runs of the same ESM. For Case 3, we wanted to apply these principles to different ESMs to quantify the differences in the apparent relationships and highlight challenges that arise in comparing relationships between ESMs. The model runs of Cases 1 and 2 using BLING as a BC afforded us the opportunity to test a "best-



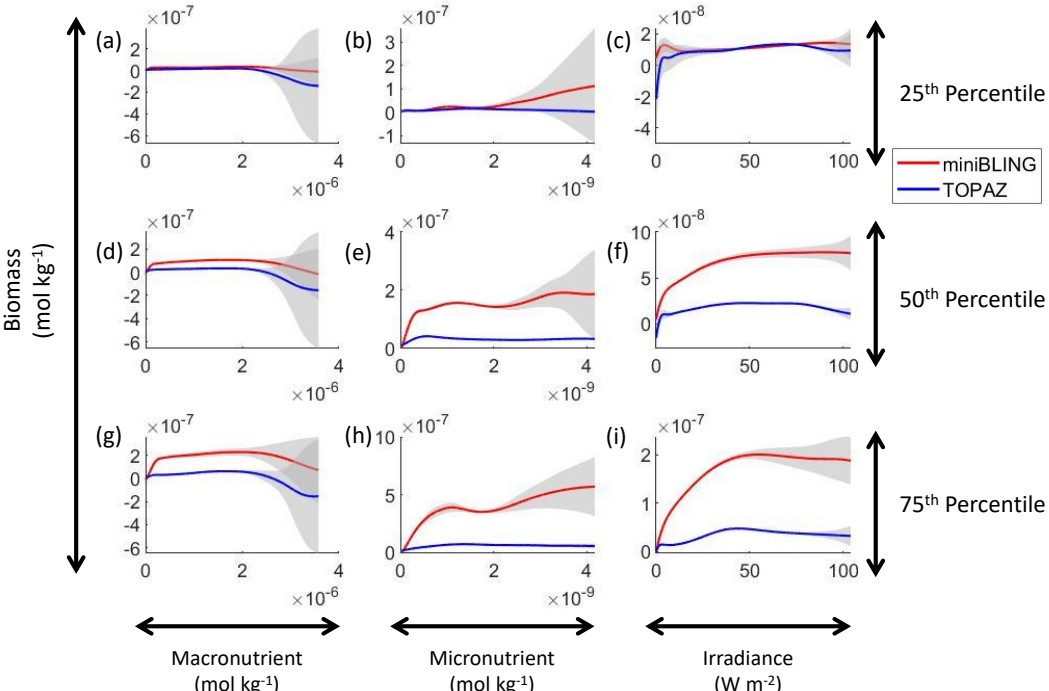

**Figure 11:** Sensitivity analysis plots for phytoplankton biomass for Case 3. Each line is the prediction for the NNE specific to each ESM and the color of each line represents the ESM (miniBLING – Red; TOPAZ – Blue). The gray region around each line shows one standard deviation in the predictions of the individual NNs that make up each NNE (ex. The gray region around the solid red curves shows the standard deviation in the predictions of the 25 NNs that make up that particular NNE). The rows correspond to the percentile value at which the other predictor variables were held constant (ex. Box (a) varies the macronutrient across its min-max range and holds the micronutrient and irradiance at their respective 25th percentile values). Columns show the x-axis variables as they vary between their min-max range. The y-axis in all subplots is the biomass concentration. Note that the biomass scale changes with each subplot.

case" scenario for predicting biomass from nutrients and light because of the tight coupling of growth rate and biomass (ie. knowing the growth rate means we know the biomass). In Case 3, the ESMs have different biogeochemical codes (ie. different biological equations) and identical physical circulations. One ESM (ESM2Mo with miniBLING as BC, referred to as miniBLING) was comparable to the BLING formulation in that the growth rate was tightly coupled with the biomass. However, the other ESM (ESM2Mo with TOPAZ as BC, referred to as TOPAZ) did not have as tight of a coupling. The TOPAZ simulation allowed biomass to be advected and diffused in the same way as nutrients, effectively making biomass a function of nutrients and physical circulation, which is more typical of ESMs and likely true in the real ocean, as well.





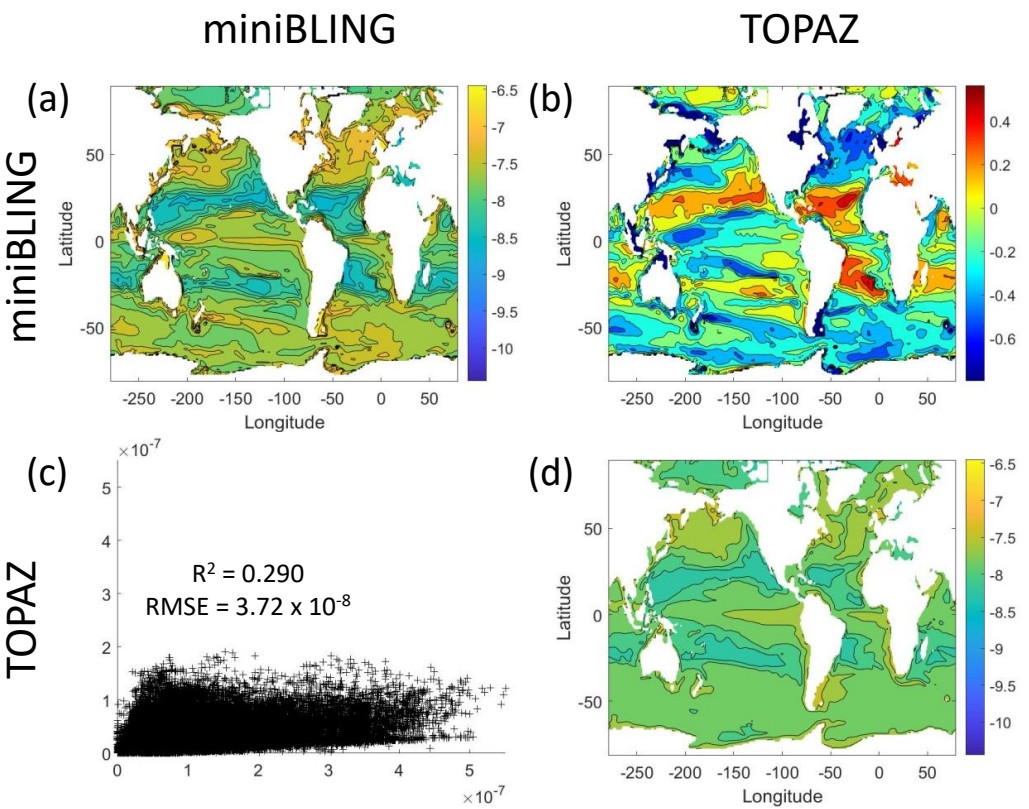

**Figure 12:** Comparison of the ESMs for total phytoplankton biomass in Case 3 in which circulation is given by ESM2Mo, but the the BCs are different. The units for biomass in all subplots are mol kg$^{-1}$. The subplots show point-by-point scatter plots comparing the ESMs against one another (c), yearly averaged log10 biomass plots for each ESM (a and d), and the log10 relative ratios comparing the yearly averaged contour plots of the ESMs (b). The x-axis and y-axis of the scatter plots (c) correspond to the horizonal/vertical ESM labels, respectively (ex. Box (c) shows the miniBLING simulation on the x-axis and the TOPAZ simulation on the y-axis). The yearly averaged log10 contour plots (a and d) correspond to the matching horizontal/vertical ESM labels (ex. Box (a) shows the yearly averaged log10 biomass of miniBLING). The log10 relative ratios (b) were calculated as the ESM listed on the horizontal axis divided by the ESM listed on the vertical axis (ex. Box (b) shows TOPAZ divided by miniBLING).


Our results indicate that the phytoplankton in the two ESMs react differently to the same conditions. It should be noted total phytoplankton biomass was used for Case 3, rather than splitting the biomass into large and small because phytoplankton output by the miniBLING BC is not differentiated into size classes. The sensitivity analysis shows that the miniBLING simulation produces higher biomass concentrations than the TOPAZ simulation under the same



**Table 5:** The performance metrics for the NNEs being used to predict the outcome of the other ESM of Case 3. In the top half of the table, the R-squared and RMSE are listed. The values in paratheses are the values from comparing the respective ESMs against one another (these are the same values listed in the respective scatter plot of Fig. 12). The values outside the parentheses are the values from using the trained NNE of the ESM listed in the row to predict the outcome of the ESM in the column (ex. The NNE trained on the TOPAZ simulation was used to predict the outcome of the miniBLING using the predictor values computed using the miniBLING simulation. These values were compared against the actual values of the miniBLING simulation to compute the RMSE of $3.91 \times 10^{-8}$). In the bottom half of the table is the percent decrease in RMSE from the number listed inside the parentheses to the RMSE outside the parentheses (a negative percent means that the error increased).

| | | | Case being predicted | |
| --- | --- | --- | --- | --- |
| | | | miniBLING | TOPAZ |
| **R-squared** | NNE being used for predicting | miniBLING | - | (0.29) 0.3985 |
| | | TOPAZ | (0.29) 0.5405 | - |
| **RMSE** | NNE being used for predicting | miniBLING | - | $(3.72 \times 10^{-8})$ $7.79 \times 10^{-8}$ |
| | | TOPAZ | $(3.72 \times 10^{-8})$ $3.91 \times 10^{-8}$ | - |
| **Percent Decrease in Error** | NNE being used for predicting | miniBLING | - | -109.29% |
| | | TOPAZ | -5.03% | - |

conditions (Fig. 11), except at lower concentrations of nutrients where they seem to react similarly (Fig. 11 a, b, c). This is not entirely unexpected since the biomass values in the miniBLING simulation were generally much higher than those in the TOPAZ simulation, as can be seen in the point-by-point comparison (Fig. 12 c). However, not all of the biomass values in the miniBLING simulation were larger than those in the TOPAZ simulation. The subtropical Atlantic regions and northern subtropical Pacific had higher yearly averaged biomass values in the TOPAZ simulation

compared to the miniBLING simulation (Fig. 12 a, b, d). As with Case 2, the additional test of asking the NNEs trained on the output of one ESM to predict the the output from the other ESM reinforced the result that different apparent relationships were found from an increase in error for both ESMs (Table 5).

The challenge of comparing the results of different ESMs was evident in Case 3. For example, the performance metrics

for the model runs in Cases 1 and 2 were relatively high in both the training and testing subsets, but the performance metrics for the TOPAZ simulation in Case 3 were much lower ($R^2 > 0.97$ vs ~0.58, respectively; Table 2). It was unclear whether this drop in performance was because we were unable to characterize the TOPAZ simulation with NNEs using predictors common to both runs or whether we simply did not include enough relevant variables. To understand this, we performed a brief analysis in which we trained NNEs on specific variables and measured their

performance with ESM output from CMIP5 ESM2M, which has TOPAZ as its BC (Table 6). One NNE was trained



using only variables that directly affected the phytoplankton growth rate (biology), one was trained using only variables that did *not* directly affect the growth rate (physics), and another was trained with both sets of variables (all). Our results indicated that we were able to characterize ESM2M (and, by extension, results produced by using TOPAZ as a BC) with NNEs with the inclusion of more relevant variables, such as nitrate, ammonium, and silicate (RMSE ~

$5.90 \times 10^{-5}$ mol N m$^{-3}$ [Table 6] vs. the average biomass value of $3.14 \times 10^{-4}$ mol N m$^{-3}$). Without the inclusion of all the relevant variables as predictors, the performance of the NNE trained on output from the TOPAZ simulation suffered compared to the NNE trained on the miniBLING simulation.

An additional challenge with comparing different ESMs is that certain variables may not be present in all ESMs. For

example, one ESM may have phosphate included as a variable and another ESM may not. This presents a problem when using the sensitivity analyses, because each NNE needs to be presented with the same conditions for direct comparability. One potential method for mitigating this could be to use proxy-variables, such that variables not common to both ESMs could be modified to represent the missing variables. For example, if one ESM had phosphate as a variable, another ESM did not, it might be possible to modify a variable that would be equivalent to phosphate,

such as nitrate. Using the Redfield ratio of 16:1 for the N:P ratio, the nitrate variable could be divided by 16 and thus be considered a proxy variable for phosphate. This proxy phosphate variable could then be used in training the NNE particular to applicable ESM, so all NNEs would be trained using the same predictors.





**Table 6:** The performance metrics for the training and testing subsets of NNEs trained on different variable combinations of CMIP5 ESM2M output and details about the predictor/target variables.

| Variable Groupings | Predictor Variables | Target Variable | Training Data | | Testing Data | |
|---|---|---|---|---|---|---|
| | | | R-squared | RMSE | R-squared | RMSE |
| All Variables | 1) Nitrate (mol m$^{-3}$)<br>2) Ammonium (mol m$^{-3}$)<br>3) Phosphate (mol m$^{-3}$)<br>4) Dissolved Iron (mol m$^{-3}$)<br>5) Silicate (mol m$^{-3}$)<br>6) Temperature (K)<br>7) Net Downward Shortwave Flux (W m$^{-2}$)<br>8) Mixed Layer Thickness (m)<br>9) Surface X-Velocity (m s$^{-1}$)<br>10) Surface Y-Velocity (m s$^{-1}$)<br>11) Upward Ocean Mass Transport at 45 m Depth (kg s$^{-1}$) | Phytoplankton Concentration (mol N m$^{-3}$) | 0.9756 | 3.61 x 10$^{-5}$ | 0.9754 | 3.65 x 10$^{-5}$ |
| Only Variables Directly Affecting Phytoplankton Growth Rate | 1) Nitrate (mol m$^{-3}$)<br>2) Ammonium (mol m$^{-3}$)<br>3) Phosphate (mol m$^{-3}$)<br>4) Dissolved Iron (mol m$^{-3}$)<br>5) Silicate (mol m$^{-3}$)<br>6) Temperature (K)<br>7) Net Downward Shortwave Flux (W m$^{-2}$) | Phytoplankton Concentration (mol N m$^{-3}$) | 0.9358 | 5.87 x 10$^{-5}$ | 0.9352 | 5.93 x 10$^{-5}$ |
| Only Variables NOT Directly Affecting Phytoplankton Growth Rate | 1) Mixed Layer Thickness (m)<br>2) Surface X-Velocity (m s$^{-1}$)<br>3) Surface Y-Velocity (m s$^{-1}$)<br>4) Upward Ocean Mass Transport at 45 m Depth (kg s$^{-1}$) | Phytoplankton Concentration (mol N m$^{-3}$) | 0.3268 | 1.90 x 10$^{-4}$ | 0.3279 | 1.91 x 10$^{-4}$ |

## 5 Conclusions

A challenge of using ESMs is understanding why different ESMs yield different results, even when they are run under
similar conditions. Our objective with this manuscript was to investigate the extent to which NNEs could characterize
differences across ESMs through differences in circulation vs differences in biological formulations. We approached
this objective by exploring three cases:

1. In the first case, we compared three configurations of an ESM that had identical intrinsic biological
relationships but different physical circulations. The purpose of this case was to quantify the extent to which
differences in physical circulations between model runs of the same ESM could affect the apparent
relationships found by NNEs.

2. In the second case, we compared two model runs from the same ESM, except that the intrinsic biological
equations were different and the physical circulations were similar. The purpose of this case was to quantify
the extent to which NNEs found differences in the apparent relationships and the size of those differences.

3. In the third case, we used two different ESMs that had different intrinsic biological relationships but identical
physical circulations. The greatest difference between them was that in one ESM (ESM2Mo with TOPAZ as



BC), biomass was able to be advected and diffused making it a function of nutrients, light *and* circulation. This was in contrast to the other ESM (ESM2Mo with miniBLING embedded as BC) where biomass was only a function of nutrients. The purpose of this case was to apply what we had learned in the first two cases to a more

realistic ESM to quantify differences in the apparent relationships and identify any challenges.

Our results indicate that when all the relevant variables are included as predictors, the NNEs are a parsimonious representation of the ESMs and we can be relatively confident in their predictions. This confidence then allows us to query these NNEs using sensitivity analyses to find the apparent relationships, which provide information on the

relationships between the predictor and target variables.

With the first case, the similar performance metrics in the within- and cross-model comparison, along with the overlapping apparent relationships demonstrated that the NNEs were able to attribute differences between the model runs to physics. Likewise, in the second case, where the biological relationships differed, the NNEs were capable of

attributing differences between the model runs to biological factors and were able to identify the elements of that formulation that were different.

With the third case, we were able to show that it is possible to compare the apparent relationships between two different ESMs and that key differences can be found. However, this case also highlighted some of the challenges when

comparing output from multiple ESMs. In order to adequately capture the variability and achieve high performance metrics, all relevant variables for predicting an outcome must be included as predictors for each NNE. However, this presents a problem when one ESM may have a variable and another ESM does not. One possible solution is to use proxy variables, such that one variable can be modified to be representative of another.

The results of our study suggest that oceanographers and climate scientists could use the methods we have demonstrated to compare apparent relationships between ESMs, in addition to using spatiotemporal distributions and time series. This is not to say that spatiotemporal information is not important; rather, the relationships and spatiotemporal information can be used to inform one another. For example, in a side-by-side comparison of contour plots between biomass and nitrate concentrations, one might expect to see high biomass in high nitrate regions.

However, if low biomass is observed in a high nitrate region, this would suggest that another factor (such as phosphate) is limiting phytoplankton growth. By visualizing the apparent relationships, one would be able to observe that phosphate has a strong limitation factor on the phytoplankton. This could then be verified with the spatial contour plot of phosphate against the original biomass and nitrate contour plots.

In addition to comparing relationships between ESMs, the methods presented here will allow for the comparison of relationships found in observational datasets to the relationships in ESMs. This will allow for better tuning of the models and more accurate representations of the natural world and what changes we might expect under climate change. For example, if the apparent relationships from observations were to indicate increased biomass with increased





CO$_2$ concentrations but current ESMs were predicting lower biomass, modelers would be able to update the ESMs
with more accurate representations or finer tuning of the parameters. We will report on these potential applications in
future work.

**Code and Data Availability**

The Matlab scripts (MATLAB, 2019) for training the NNEs and constructing the tables and figures, along with the
ESM outputs used for each case are available in the Zenodo repository (https://doi.org/10.5281/zenodo.4774438,
Holder et al. 2021).

**Author contribution**

The conceptualization and the methodology of the research was developed by CH and AG. The coding scripts that
configured the data for training, trained the NNEs, and produced the tables/figures were written by CH. The analysis
and interpretation of the data was carried out by CH and AG. AG and MAP ran the ESMs that produced the model
output for Cases 1 and 2. MAP coded the LgSm version of the BLING BC used in Case 2. The original draft of the
manuscript was written by CH and AG, with edits, suggestions, and revisions provided by MAP.

**Competing Interests**

The authors declare that they have no conflicts of interest.

**Financial Support**

This research was supported by the National Science Foundation (NSF) Division of Ocean Sciences (OCE) (Grant
No. 1756568) and the Department of Energy (Grant No. DE-SC0019344).



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
