# Peer review of "Using Neural Network Ensembles to Separate Ocean Biogeochemical and Physical Drivers of Phytoplankton Biogeography in Earth System Models"

_Geoscientific Model Development, 2021_

## Referee Comment (RC1)

**Review of "Using Neural Network Ensembles to Separate Biogeochemical and Physical Components in Earth System Models" by Holder et al. (gmd-2021-167)**

October 8, 2021

In their article, Holder et al. use the approach of neural network ensembles (NNE) to extract relationships between predictor (nutrients, irradiance, temperature) and target (small and large phytoplankton biomass) variables within ocean biogeochemical models. Specifically, they investigate whether the NNE approach is capable of determining why different models produce different results. They study three test cases, where they either alter the physical formulation controlling the circulation or biological equations. Thereby, they focus on the two different types of relationship, i.e., intrinsic vs. apparent relationships. They conclude that the NNE approach is capable of characterizing these relationships and can thus be considered as a parsimonious representation of the system, including extrapolative power.

Overall, this study provides a valuable contribution of how one can leverage "Machine Learning" approaches to better understand the dynamics of a complex model, such as ocean biogeochemical or Earth system models. Also, not being an expert in ocean biogeochemical modeling, I consider the presented methods and analyses to be robust. The manuscript is well written, however some passages need restructuring and the manuscript needs to be tuned for its target audience — please see my comments.

I recommend minor revisions of the manuscript before publication.

**1 General Comments:**

1.1 *Overall, the manuscript based on the title and the bigger part of the abstract aims at a larger readership working with Earth system models (ESM), the main part of the manuscript is, however, very focused on ocean biogeochemistry modeling. For example, the abstract is very ESM-general until line 12, but then jumps into a very specific problem on phytoplankton. Earth system modelers, who are not so familiar with ocean biogeochemistry, might be a bit lost here and in general throughout the article. I suggest to either sharpen the focus of the manuscript to only aim for the ocean biogeochemistry community, or to be more inclusive for Earth system modeler in general. The latter solution would require that you clearly state that the ocean biogeochemistry problem investigated in this study is used as a case study to demonstrate your approach, introduce the reader more to the problem of small vs. large phytoplankton prediction, and how one could adapt your approach/case study to other aspects of the Earth system.*

**2 Specific Comments:**

2.1 *Please stick to the tenses, i.e. do not switch between present and past tense when describing your results. I recommend that you always use present tense when describing your study at hand, i.e. when describing your methods, your results etc., and only use past tense when referring to already published studies.*

2.2 *L1: The title is too general. There are also biogeochemical and physical components in the land-surface models. Better to add "Ocean" in the title.*

2.3 *L22: The abstract misses a concluding sentence. Please add a sentence that gives a general outlook of your study and highlights its significance for the discipline of Earth system modeling.*

2.4 *L27: It is limited not only by imperfect knowledge, but also by the fact that we cannot resolve the processes in current models and current HPC facilities.*

2.5 *L46: Maybe better "are indeed being modelled".*

2.6 *L50: Include a sentence here that shortly explains the concept behind the NNE.*

2.7 *L51: Again, the use of tenses in this manuscript is a bit misleading. It is better to write: "... (NNEs) **are** able to extract ..." instead of "... were able ...". It's not that they lost the capability to do so in the meantime.*

2.8 *L64: Better "high irradiance" instead of "high light".*

2.9 *L71: The paper is, on the one hand, specific about ocean biogeochemical modeling and, on the other hand, it tries to be more general about Earth system modeling. One could add a statement here that you look into phytoplankton physiology as a case study, but the approach is also applicable to other problems in the Earth system.*

2.10 *L73–83: This section reads a bit like you already discuss your results. It would work better if you used present tense and explain the different approaches which are applied in this research, and why.*

2.11 *L110: "**ocean** biogeochemical components of ESMs"*

2.12 *L112; Equation 1: Please replace "Light" with $I$ for irradiance.*

2.13 *L131: "computationally cheap".*

2.14 *L132: either "in" or "within" the model.*

2.15 *Case Descriptions: Could you include for each case an equation describing how the NN is set up? E.g. something like Biomass = NN(Irradiance, Nutrients, Temperature) with proper variable names?*

2.16 *L190: Can you more clearly explain what the "LgSm" acronym is referring to?*

2.17 *L231: "NNEs possess some capability of extrapolating outside the range of the data on which they are trained." Very important point - you should provide a citation here!*

*2.18 L232: With RF you mean Random Forests, I assume. Can you make it clear?*

*2.19 L248–250: Why did you not set up your NN system with training, test and validation datasets? So, validation dataset to prevent overfitting, and test dataset to test generalizability?*

*2.20 L260: Maybe you can write that hyperparameters tuning showed that the setup is not very sensitive to the selection of different hyperparameters.*

*2.21 L262: Did you also use a different scheme for normalization, e.g. normalization to zero mean and unit standard deviation.*

*2.22 L340: For me, the extrapolative power of your NNE approach is a very encouraging result. You show that a NN can learn the dynamics of the system from the PI run and is able to extrapolate to extrem forcing like 4xCO2 - maybe one should make a bigger deal out of this and highlight in the abstract.*

*2.23 L437: You have not introduced the abbreviations Chl:C. I know, it is clear for the reader with ocean biogeochemistry background, but your title addresses a larger readership. So, please introduce all abbreviations.*

*2.24 L498: Better rename to "Summary & Conclusions".*

*2.25 L518: Rephrase "we can be relatively confident" to something like "their predictions can be considered reliable."*

*2.26 Conclusions: Overall, I find them too long and not to-the-point. Can you boil it down to a few concise statements?*

*2.27 Figure 1 & 2: I cannot comment on the specifics of the ocean biogeochemical models. Ideally, another referee with the needed expertise should comment on these aspects.*

*2.28 Sensitivity Analysis Figures: The colored lines are the actual model run output, right? Or is it the mean NNE? The grey shading is the NNE, right? Could you put this in the legend? If the actual model output is not included in the figure, where do you show the performance for NNE versus actual model output except $R^2$ and RMSE values in the tables.*

*2.29 Sensitivity Analysis Figures: Why do you show in e.g. Figure 3 small and large phytoplankton biomass together and Figure 4 only small phytoplankton biomass. Can you not remove small phytoplankton biomass from Figure 3 and corresponding subsequent figures?*

*2.30 Sensitivity Analysis Figures: I find the black arrows at the axis to be a bit misleading - do you need them?*

*2.31 Sensitivity Analysis Figures: What does "ex." in the captions mean? Example? Better use e.g. then.*

*2.32 Figure 5: I'd prefer if you added the unit next to the colorbar.*

---

## Referee Comment (RC2)

**Review for "***Using Neural Network Ensembles to Separate Biogeochemical and Physical Components in Earth System Models***" by Holder et al. (gmd-2021-167)**

14[th] Nov. 2021

**SUMMARY**

The authors' present an efficient and innovative approach for identifying the major sources of difference between Earth System Model (ESM) predictions. Specifically, they use neural network ensembles (NNEs) to identify whether the phytoplankton biomass predictions of different ESMs are more affected by changes in ocean circulation, or by differences in biogeochemical formulation. They conclude that, in the context of their test cases, the NNEs were able to accurately identify the relationships between variables in the ESMs – when they have access to all of the variables that affect phytoplankton biomass.

On the whole, the authors' have developed a robust, well-designed and meticulously implemented framework for examining variability in the outputs of ESMs. Their NNEs appear to perform exceptionally well, and serve as a powerful demonstration of the predictive capabilities of such models. The manuscript is generally well-written, and, in the context of their initially stated aims, the authors' have done an excellent job.

That said, there are areas where I feel this paper could be improved. I recommend minor revisions before publication, primarily in terms of ensuring that the motivation of the study and the broader importance of the results are both clearly communicated.

**GENERAL COMMENTS**

**1.1 *Main motivations for study unclear from abstract and intro**

I feel that the Abstract and Introduction could do a better job of presenting the work in terms of the specific problems that the authors' methods are addressing, and why it matters. They touch on several factors that might lead different ESMs to produce different outputs, but it is not necessarily clear, in my opinion, how the work being introduced addresses these problems.

- Different ESMs yield different predictions because of variations and uncertainties in input parameters (line 36-37)
- Uncertainty as to whether ESMs are using the "correct relationships" (line 45-46)
- Traditional methods for estimating ESM sensitivity are inadequate (line 40-44)

Thereafter, it's concluded that these factors indicate a need for a method that can identify whether different ESM predictions of phytoplankton biomass result from differences in biogeochemical formulation, or in physical circulations (line 47-49). It's not immediately clear how the proposed method will help alleviate the previously raised issues.

**1.2 *Clarify intended audience**

- Who is the intended audience?
- Researchers who develop ESMs? If so, how will your methods/results help improve their models?
- Researchers who work with observational data? How might this work help them better utilise their data?
- Researchers who build ML models? How might this work inform theirs? etc

The conclusion does elaborate on some of the reasons why other researchers (ostensibly those who wish to compare different ESMs) might find value in this work, and reads more clearly than the abstract and introduction, if a little unfocused.

**1.3 *Clarify broader importance**

The conclusion does briefly elaborate on some of the reasons why other researchers might find value in this work. Is the primary target audience those who wish to compare the outputs of different ESMs? Those who wish to improve their existing ESM? Those who wish to more efficiently utilize observational data?

In NNEs, the authors' demonstrate a very powerful tool, that can (and is being) applied to all of the above use cases across multiple fields. That said, the application of this methodology in the current study is quite specific to extracting relationships from ESMs. However, the authors' suggest that directly applying their methods to observational data will help calibrate and improve ESMs, and thus yield better predictions of e.g. changes we might expect under climate change.

This is a big claim to make, and I question whether it is meaningful in the context of this manuscript. In the current work, the authors' have access to a complete, perfect knowledge of all of the variables that affect e.g. plankton biomass, within each ESM (itself a highly simplified representation of the real Earth system). In addition, their NNEs have access to ALL the depth-integrated data in every part of the simulated global ocean for each ESM, across arbitrary time.

Real world data clearly represents a very different set of challenges and constraints. Observational datasets are orders of magnitude more sparse and imbalanced. Even if we were able to sample the entire ocean, our knowledge of the important physical and chemical fluxes driving growth and distribution is incomplete, even without including the significant added complexities of biotic interactions and adaptive evolution.

The results of the present work do not appear to be sufficient to make claims as to the direct applicability of these methods, as presently described, to generating more accurate representations of the natural world.

**SPECIFIC COMMENTS**

Line 11-14 – Is this really a "compensating error"? In the real ocean, oligotrophic regions are dominated by small phytoplankton species (greater surface-area-to-volume ratio, greater uptake of, and thus "sensitivity to", nutrients)

Line 14-15 - Are the authors' referencing their own previous work here? If so, this should be made clear: "Recently, we demonstrated that... "

Line 17-18 - Suggest being more specific about the types of results being examined e.g. "...why different ESMs produce different spatiotemporal distributions of phytoplankton biomass"

Line 20 - Three test cases are mentioned, but only two are elaborated in the abstract.

Line 28 - Yes, but also by the increasingly prohibitive computational expense of adding complexity and resolution.

Line 33-35 - This seems like a reasonable metric to vary, particularly when modelling different plankton community structures. It is not necessarily variable as a result of uncertainty.

Line 36-49 – This paragraph first mentions the uncertainty associated with ESM input parameters, then the coupled nature of a given input to multiple outputs, and then the difficulty in knowing whether ESMs are modelling the "correct relationships". These are all valid – if separate – points. But I'm struggling to link these points to the proposed 'solution' in lines 47-49.  Will the NNE help to identify which relationships are 'most correct', or extract new 'more correct' relationships? Or is its primary function to more clearly identify the reasons why ESM predictions of biological variables diverge?

Line 56-59 - This definition was a little confusing to read. Are the "intrinsic relationships" those which are known as true drivers of a target variable? E.g. those captured by lab growth rate experiments, or, as in the current context, the biogeochemical equations underlying ESMs?

Line 61-64 - Similar to the previous point, are your "apparent relationships" a reference to data-derived correlations?

For the record, I really like the terms "intrinsic" and "apparent", but I think your description of these terms is much more clear in your previous work "Can machine learning extract the mechanisms controlling phytoplankton growth from large-scale observations?".

Line 71-72 – Perhaps 'determining *the most significant sources of* differences in ESM outputs'?

Line 73 – Can '*combinations of these two*' be considered as an independent 'primary driver'?

Line 78-79 - Possibly worth clarifying that you're only referring to the link between circulation changes and patterns of co-limitation in the ESM (to avoid readers' potentially drawing parallels to real ocean dynamics, to which such findings may not apply).

Line 97 – A reader's question here might be - "why not identical?"

Line 162 - For all case descriptions, perhaps include details on e.g. how long each ESM was run for (in model years), output formats (e.g. daily/monthly averaged values) and the model resolution.

Line 166-167 - Perhaps expand on this, as it seems like an important point. We know from your (very clear and helpful!) Fig. 1 that nutrient distribution is coupled to circulation, but biomass itself is not, and that changes in biomass are a function only of nutrient distribution. With this in mind, the reader might be wondering whether it is even possible – given the constraints of BLING - to "push the biology into fundamentally new states" by varying circulation alone.

Line 236-237 - Why were these activation functions chosen?

Line 257-258 - Perhaps mention these previous sensitivity tests earlier (e.g. line 235+) - "we previously determined that {x,y} were not sensitive to {p,q} (ref) so our individual NN's were constructed using…"

Line 274-275 - I think it's worth including more detail on the actual data that the NNEs are using. How many datapoints do the training and test sets contain? Are the training and test sets randomly sampled in both time and space from the ESM outputs? Are they drawn from different temporal periods? Or from different spatial regions?

Were any resampling techniques employed to address potential imbalance in the randomly-sampled data? Or were the datasets large enough to effectively capture variance? Did you use all of the depth-integrated output data from the model runs for training/testing?

Line 281-282 – I'd suggest being more explicit here on how r-squared was calculated - this is a notoriously tricky, often misused metric. E.g. NNE predictions of mean annual biomass for each point are plotted against the 'true' ESM values… standard or adjusted, etc.

Line 290-294 – What were your criteria for what constituted a significant increase or decrease in RMSE?

Line 320-321 - I suspect that some readers will have questions about the extremely high performance seen here, across both metrics. It would be helpful again to provide more detail on the nature of the training and test datasets, how they were sampled, etc. Is the "mean value of the total biomass" calculated as a total global mean, or the mean for a given point? Across what time period?

Line 324-328 - Is this unexpected? The fact that "physical circulation would simply act to change the location of where combinations of light and nutrients were found" seems like a given, considering that "biomass is not directly affected by changes in the physical circulation" in BLING.I think that an explicit clarification of the importance of this result, in this context, would be helpful to place here.

Line 483-485 - This is a really interesting result in terms of the importance of including the correct variables in predictive models. In this case, we happen to know all of the variables that affect our target within the system. When applying such models to real-world data, we don't, and it can have significant consequences for predictive accuracy.

**TECHNICAL COMMENTS**

Line 1-2 - The use of "components" in the title is a little broad. Perhaps substitute with "Drivers of Plankton Biogeography"?

Line 92-94 - Advise being more specific here - what is meant by "push the biology into fundamentally new states"? I liked "produce new patterns of colimitation", as given in line 79.

Line 110 – Introduce symbol for phytoplankton biomass (B) in this line

Line 138 - Typo, should read "than the"

Line 142 - Typo "nutrient and temperature"

Line 180 – Is this meant to read 'Section 3.4'?

Line 311 - Again, I think your original phrase "new patterns of colimitation" is more descriptive and appropriate than "fundamentally new states"

Fig. 3 and 4 - Agreed, the inclusion of both large and small phytoplankton in Fig 3 makes it difficult to read. Suggest splitting them up

---

## Author Comment (AC1)

*Author Responses Addressing Review from Referee #1 for* "Using Neural Network Ensembles to Separate Biogeochemical and Physical Components in Earth System Models" *by* Holder et al.

For these responses, we address each Referee comment individually and include our response below it. The Referee Comments (RC) are numbered and use a black font, while the Author Responses (AR) use a red font.

RC0.0: In their article, Holder et al. use the approach of neural network ensembles (NNE) to extract relationships between predictor (nutrients, irradiance, temperature) and target (small and large phytoplankton biomass) variables within ocean biogeochemical models. Specifically, they investigate whether the NNE approach is capable of determining why different models produce different results. They study three test cases, where they either alter the physical formulation controlling the circulation or biological equations. Thereby, they focus on the two different types of relationship, i.e., intrinsic vs. apparent relationships. They conclude that the NNE approach is capable of characterizing these relationships and can thus be considered as a parsimonious representation of the system, including extrapolative power.

Overall, this study provides a valuable contribution of how one can leverage "Machine Learning" approaches to better understand the dynamics of a complex model, such as ocean biogeochemical or Earth system models. Also, not being an expert in ocean biogeochemical modeling, I consider the presented methods and analyses to be robust. The manuscript is well written, however some passages need restructuring and the manuscript needs to be tuned for its target audience — please see my comments.

I recommend minor revisions of the manuscript before publication.

AR0.0: We want to thank Referee 1 for their helpful comments and suggestions. We have done our best to address each of the comments below.

RC1.1: Overall, the manuscript based on the title and the bigger part of the abstract aims at a larger readership working with Earth system models (ESM), the main part of the manuscript is, however, very focused on ocean biogeochemistry modeling. For example, the abstract is very ESM-general until line 12, but then jumps into a very specific problem on phytoplankton. Earth system modelers, who are not so familiar with ocean biogeochemistry, might be a bit lost here and in general throughout the article. I suggest to either sharpen the focus of the manuscript to only aim for the ocean biogeochemistry community, or to be more inclusive for Earth system modeler in general. The latter solution would require that you clearly state that the ocean biogeochemistry problem investigated in this study is used as a case study to demonstrate your approach, introduce the reader more to the problem of small vs. large phytoplankton prediction, and how one could adapt your approach/case study to other aspects of the Earth system.

AR1.1: In the updated manuscript, we specify that although we focus on phytoplankton and ocean components that our results are applicable to other components of ESMs as well.

RC2.1: Please stick to the tenses, i.e. do not switch between present and past tense when describing your results. I recommend that you always use present tense when describing your study at hand, i.e. when describing your methods, your results etc., and only use past tense when referring to already published studies.

AR2.1: This was a very helpful comment! In the updated manuscript, we replaced past tenses with present tenses in the following sections: Methods, Case Descriptions, Results and Discussion. We replaced present tenses in the Conclusions section with past tenses.

RC2.2: L1: The title is too general. There are also biogeochemical and physical components in the land-surface models. Better to add "Ocean" in the title.

AR2.2: Added "Ocean" into the title for clarity.

RC2.3: L22: The abstract misses a concluding sentence. Please add a sentence that gives a general outlook of your study and highlights its significance for the discipline of Earth system modeling.

AR2.3: Added two short concluding sentences to the end of the abstract in the updated manuscript.

RC2.4: L27: It is limited not only by imperfect knowledge, but also by the fact that we cannot resolve the processes in current models and current HPC facilities.

AR2.4:  Updated in the revised manuscript.

RC2.5: L46: Maybe better "are indeed being modelled".

AR2.5: Updated with the suggested sentence fragment.

RC2.6: L50: Include a sentence here that shortly explains the concept behind the NNE.

AR2.6: Added a transition sentence at the bottom of the paragraph ending in "… differences in physical circulations and climate sensitivities." Also added a sentence to the paragraph following the one previously mentioned which briefly introduces and explains the concept behind NNEs.

RC2.7: L51: Again, the use of tenses in this manuscript is a bit misleading. It is better to write: "… (NNEs) **are** able to extract …" instead of "… were able …". It's not that they lost the capability to do so in the meantime.

AR2.7: We address this particular comment as part of the RC2.1 comment above.

RC2.8: L64: Better "high irradiance" instead of "high light".

AR2.8: Updated with the suggested wording.

RC2.9: L71: The paper is, on the one hand, specific about ocean biogeochemical modeling and, on the other hand, it tries to be more general about Earth system modeling. One could add a statement here that you look into phytoplankton physiology as a case study, but the approach is also applicable to other problems in the Earth system.

AR2.9: We added a sentence specifying that this approach is applicable to other components of ESMs and specified that we focus on marine phytoplankton physiology in our study.

RC2.10: L73–83: This section reads a bit like you already discuss your results. It would work better if you used present tense and explain the different approaches which are applied in this research, and why.

AR2.10: We changed the wording of these sentences to use the present tense, instead of past tense, so that we are not discussing the results of our study in the introduction.

RC2.11: L110: "**ocean** biogeochemical components of ESMs"

AR2.11: Updated with the suggested wording.

RC2.12: L112; Equation 1: Please replace "Light" with **I** for irradiance.

AR2.12: Replaced "light" with irradiance in Equation 1 and in the sentence describing the variables of Equation 1. Also replaced the term "light" with "irradiance" throughout the rest of the text as well, including the abbreviations in the equations.

RC2.13: L131: "computationally cheap".

AR2.13: Updated with the suggested wording.

RC2.14: L132: either "in" or "within" the model.

AR2.14: Removed "in" and kept "within."

RC2.15: Case Descriptions: Could you include for each case an equation describing how the NN is set up? E.g. something like Biomass = NN(Irradiance, Nutrients, Temperature) with proper variable names?

AR2.15: We tried to implement this suggestion where we gave each Case (3.1, 3.2, and 3.3) their own equation, but this led to a lot of equations repeating themselves. Additionally, we wanted to keep most details pertaining to the framework of the NNs and NNEs in Section 3.4 for clarity.

To include the type of information requested in this comment, at the end of Section 3.1 we added a sentence stating that the details of the NNE and NN training/frameworks can be found Section 3.4. Additionally, in Section 3.4 we added an updated description for the structure of the individual NNs.

RC2.16: L190: Can you more clearly explain what the "LgSm" acronym is referring to?

AR2.16: The LgSm acronym was chosen because there are variables that specifically state the concentration of the small and large phytoplankton biomass. This differs from the PI Control where the small and large phytoplankton biomass are calculated as a fraction of the total biomass. In both instances, you still get small and large phytoplankton biomass values, but just arrive at it slightly differently. Specifying the acronym as LgSm is shorthand for stating that the small/large phytoplankton variables are specifically stated in that model run.

RC2.17: L231: "NNEs possess some capability of extrapolating outside the range of the data on which they are trained." Very important point - you should provide a citation here!

AR2.17: Included another mention of Holder and Gnanadesikan (2021) since that is something that particular study found.

RC2.18: L232: With RF you mean Random Forests, I assume. Can you make it clear?

AR2.18: Replaced the RF acronym with the full spelling of random forests.

RC2.19: L248–250: Why did you not set up your NN system with training, test and validation datasets? So, validation dataset to prevent overtraining, and test dataset to test generalizability?

AR2.19: The Matlab function that we used for training the individual NNs separates the data into training, validation, and test datasets. We were trying to keep the specific details in the manuscript to a minimum, but we understand the need for this clarification. For clarity, we have included details about this in the updated manuscript.

We can also provide a brief explanation here and please note the specific distinction we make between *dataset* and *subset*. In the original manuscript, we mention that we split the data into training and testing datasets. Only the training dataset is provided to the Matlab function used for training the NNs. The Matlab function then takes the training dataset and splits it further into training, validation, and testing *subsets*, with 70% of the data from the training dataset going into the training *subset*, 15% to the validation subset, and 15% to the testing subset. The remaining observations in the testing *dataset* are therefore observations that none of the trained NNs have ever seen before, which makes the performance metrics even more rigorous. This provides a convenient way to test the unique performance of the NNE (collection of the trained NNs).

RC2.20: L260: Maybe you can write that hyperparameters tuning showed that the setup is not very sensitive to the selection of different hyperparameters.

AR2.20: Depending on the hyperparameters, the performance of the NNEs could be affected. For example, if the neural networks used hidden layers that had only one node or that used a linear activation function, the performance would decrease. For clarity, we have changed this paragraph to include more specific information.

RC2.21: L262: Did you also use a different scheme for normalization, e.g. normalization to zero mean and unit standard deviation.

AR2.21: We considered normalizing with the zero mean and unit standard deviation, but the predictors are either heavily right-skewed (nutrients) or bimodal (temperature). Even with a different normalization scheme, we still get values greater than 3 standard deviations from a zero mean.

We could include this normalization before scaling the variables between -1 and 1, but we already get relatively short training times for the NNs with the current parameters.

RC2.22: L340: For me, the extrapolative power of your NNE approach is a very encouraging result. You show that a NN can learn the dynamics of the system from the PI run and is able to extrapolate to extreme forcing like 4xCO2 - maybe one should make a bigger deal out of this and highlight in the abstract.

AR2.22: Yes, it is an encouraging result, but we were trying to be careful about how we stated this result. We did not want to state that this method is great for extrapolating. Using any method for extrapolation comes with higher uncertainty in the regions of the dataspace where the model was not trained. For example, NNEs will have higher uncertainty in the regions where all the predictor variables are very high, because there are not any observations from that region of the dataspace in the training subset. Any predictions the NNEs make in that unexplored region will be less certain than regions of the dataspace that were included in the training subset.

One way to explain the predictability of the 4xCO2 from the NNE trained on the PI Control run is that the PI Control run and the 4xCO2 are being governed by the same equations. Although they have different inputs, the models are still run with the same internal equations and constants. If one of the constants (e.g., different half-saturation constant for one of the nutrients) between the two runs differed, the apparent relationships would be different, and the accuracy of the predictions would decrease when using one NNE to predict the outcome of the other. In the original version of the manuscript, we state this in Lines 378-379, "When the biological equations remain the same, changing the physical parameters simply change where combinations of nutrients and light occur." To make this point more apparent, we have added an additional sentence clarifying this in the final paragraph of Section 4.1 in the updated manuscript.

RC2.23: L437: You have not introduced the abbreviations Chl:C. I know, it is clear for the reader with ocean biogeochemistry background, but your title addresses a larger readership. So, please introduce all abbreviations.

AR2.23: Defined the acronym.

RC2.24: L498: Better rename to "Summary & Conclusions".

AR2.24: Updated the section name to the suggested wording.

RC2.25: L518: Rephrase "we can be relatively confident" to something like "their predictions can be considered reliable."

AR2.25: We agree the current wording could be improved. We have revised this in the updated manuscript.

RC2.26: Conclusions: Overall, I find them too long and not to-the-point. Can you boil it down to a few concise statements?

AR2.26: In the updated manuscript, we have shortened the conclusions to what we consider to be the essential points that we wanted to highlight. We kept the summary of each case (L499-L515) to remind readers of the main objectives for each one. We condensed the next three paragraphs (L517-L533) into a single paragraph to summarize the main conclusions. We kept the next two paragraphs (L535-L551) which discuss the implications of the research and how the research can be utilized by oceanographers and climate scientists.

RC2.27: Figure 1 & 2: I cannot comment on the specifics of the ocean biogeochemical models. Ideally, another referee with the needed expertise should comment on these aspects.

AR2.27: Understood and noted.

RC2.28: Sensitivity Analysis Figures: The colored lines are the actual model run output, right? Or is it the mean NNE? The grey shading is the NNE, right? Could you put this in the legend? If the actual model output is not included in the figure, where do you show the performance for NNE versus actual model output except $R^2$ and RMSE values in the tables.

AR2.28: The colored lines are the average of the predictions from the NNs that make up the respective NNE. They grey shading is equivalent to plus/minus one standard deviation relative to the predictions of those NNs. In the updated manuscript, we have kept the legend the same in order to minimize the space required for the legend. However, we have updated the description of each sensitivity analysis figure to make it clear that the lines are the average prediction of the NNEs.

The actual relationship of the model is not included since the model output does not have that capability. One purpose of the apparent relationships is to allow for the visualization of those relationships. The proof-of-concept for using the apparent relationships in this way is discussed in Holder and Gnanadesikan (2021). Within that manuscript the actual model output is shown, along with the predictions from several machine learning methods.

RC2.29: Sensitivity Analysis Figures: Why do you show in e.g. Figure 3 small and large phytoplankton biomass together and Figure 4 only small phytoplankton biomass. Can you not remove small phytoplankton biomass from Figure 3 and corresponding subsequent figures?

AR2.29: We included Figure 4 with only the small phytoplankton biomass since the small phytoplankton lines are overshadowed by the responses of the large phytoplankton in the higher

percentiles of Figure 3, such as the 75th percentile nutrient and temperature subplots. The benefit of including them both on the same original plot is that it allows for the comparison of the apparent relationships between small and large phytoplankton. Since the large phytoplankton relationships are clear in all the subplots of Figure 3, we did not think it was necessary to create a separate figure for large phytoplankton like we did for small phytoplankton in Figure 4.

RC2.30: Sensitivity Analysis Figures: I find the black arrows at the axis to be a bit misleading - do you need them?

AR2.30: We understand how they can be misleading. The only black arrows we kept were the ones labeling the biomass, so that it is obvious that the y-axis on each plot is for biomass. The rest of the black arrows are not necessary for communicating the purpose of the figures and we have removed them in the updated manuscript.

RC2.31: Sensitivity Analysis Figures: What does "ex." in the captions mean? Example? Better use e.g. then.

AR2.31: Yes, "ex" was being used as shorthand for "for example." We have updated all instances of "ex" with "e.g." in the updated manuscript.

RC2.32: Figure 5: I'd prefer if you added the unit next to the colorbar.

AR2.32: Label and units added to the colorbars of the contour plots.

---

## Author Comment (AC3)

*Author Responses Addressing Review from Referee #2 for* "Using Neural Network Ensembles to Separate Biogeochemical and Physical Components in Earth System Models" *by* Holder et al.

For these responses, we address each Referee comment individually and include our response below it. The Referee Comments (RC) are numbered and use a black font, while the Author Responses (AR) are also numbered and use a red font.

RC0.0: The authors' present an efficient and innovative approach for identifying the major sources of difference between Earth System Model (ESM) predictions. Specifically, they use neural network ensembles (NNEs) to identify whether the phytoplankton biomass predictions of different ESMs are more affected by changes in ocean circulation, or by differences in biogeochemical formulation. They conclude that, in the context of their test cases, the NNEs were able to accurately identify the relationships between variables in the ESMs – when they have access to all of the variables that affect phytoplankton biomass.

On the whole, the authors' have developed a robust, well-designed and meticulously implemented framework for examining variability in the outputs of ESMs. Their NNEs appear to perform exceptionally well, and serve as a powerful demonstration of the predictive capabilities of such models. The manuscript is generally well-written, and, in the context of their initially stated aims, the authors' have done an excellent job.

That said, there are areas where I feel this paper could be improved. I recommend minor revisions before publication, primarily in terms of ensuring that the motivation of the study and the broader importance of the results are both clearly communicated.

AR0.0: We want to thank Referee 2 for their helpful comments and suggestions. We have done our best to address each of the comments below.

RC1.1: **Main motivations for study unclear from abstract and intro**

I feel that the Abstract and Introduction could do a better job of presenting the work in terms of the specific problems that the authors' methods are addressing, and why it matters. They touch on several factors that might lead different ESMs to produce different outputs, but it is not necessarily clear, in my opinion, how the work being introduced addresses these problems.

- Different ESMs yield different predictions because of variations and uncertainties in input parameters (line 36-37)
- Uncertainty as to whether ESMs are using the "correct relationships" (line 45-46)
- Traditional methods for estimating ESM sensitivity are inadequate (line 40-44)

Thereafter, it's concluded that these factors indicate a need for a method that can identify whether different ESM predictions of phytoplankton biomass result from differences in biogeochemical formulation, or in physical circulations (line 47-49). It's not immediately clear how the proposed method will help alleviate the previously raised issues.

AR1.1: We have updated the portion of the introduction that you reference in the updated manuscript. In particular, we more clearly state the objective of the paper earlier and mention NNEs as the method we use in the paper to help us achieve that objective.

RC1.2: **Clarify intended audience**

- Who is the intended audience?

- Researchers who develop ESMs? If so, how will your methods/results help improve their models?
- Researchers who work with observational data? How might this work help them better utilise their data?
- Researchers who build ML models? How might this work inform theirs? etc

The conclusion does elaborate on some of the reasons why other researchers (ostensibly those who wish to compare different ESMs) might find value in this work, and reads more clearly than the abstract and introduction, if a little unfocused.

AR1.2: In the updated manuscript, we have narrowed the focus in terms of the specific audience. We also state that this method is applicable to other research areas and other components of ESMs, although we only focus on marine phytoplankton in our paper.

RC1.3: **Clarify broader importance**

The conclusion does briefly elaborate on some of the reasons why other researchers might find value in this work. Is the primary target audience those who wish to compare the outputs of different ESMs? Those who wish to improve their existing ESM? Those who wish to more efficiently utilize observational data?

In NNEs, the authors' demonstrate a very powerful tool, that can (and is being) applied to all of the above use cases across multiple fields. That said, the application of this methodology in the current study is quite specific to extracting relationships from ESMs. However, the authors' suggest that directly applying their methods to observational data will help calibrate and improve ESMs, and thus yield better predictions of e.g. changes we might expect under climate change.

This is a big claim to make, and I question whether it is meaningful in the context of this manuscript. In the current work, the authors' have access to a complete, perfect knowledge of all of the variables that affect e.g. plankton biomass, within each ESM (itself a highly simplified representation of the real Earth system). In addition, their NNEs have access to ALL the depth-integrated data in every part of the simulated global ocean for each ESM, across arbitrary time.

Real world data clearly represents a very different set of challenges and constraints. Observational datasets are orders of magnitude more sparse and imbalanced. Even if we were able to sample the entire ocean, our knowledge of the important physical and chemical fluxes driving growth and distribution is incomplete, even without including the significant added complexities of biotic interactions and adaptive evolution.

The results of the present work do not appear to be sufficient to make claims as to the direct applicability of these methods, as presently described, to generating more accurate representations of the natural world.

AR1.3: Similar to AR1.2, our primary audience is modellers, but we also briefly discuss how these methods can be applied to other oceanographic datasets and why they might be of interest to other Earth scientists.

Although it is a big claim to state that these methods can be applied to observations, it is based on the work of another manuscript we are currently working on. In that manuscript, we demonstrate that using *climatologies* of ESM outputs and interpolated *climatologies* of observations from various data sources, we can compare the two. For example, using sensitivity analyses we can examine the general trend in the apparent relationships for various ESMs and how they compare to the trend in observations. Additionally, our preliminary results suggest that we can capture a large portion of the variance in climatological observational datasets with machine learning (60-80%). Although

observations are certainly more sparse and imbalanced, using climatologies we can make a "first-pass" of the comparison between the two.

RC2.1: Line 11-14 – Is this really a "compensating error"? In the real ocean, oligotrophic regions are dominated by small phytoplankton species (greater surface-area-to-volume ratio, greater uptake of, and thus "sensitivity to", nutrients)

AR2.1: The example of weak upwelling, low nutrients, and nutrient sensitivity was only meant to serve as an example of something that an ESM **could** do to possibly compensate. The main point we were trying to describe is that the output of ESMs (such as spatiotemporal distributions) might match observations for the wrong reasons, e.g., incorrect assumptions, equations, etc. Different configurations of an ESM can arrive at the same answer for very different reasons. This means that just because the output of an ESM contour map matches a contour map of observations, the ESM might have arrived at the correct distribution for a reason other what actually happened in the real world. We have updated the wording in the updated manuscript to better reflect this.

RC2.2: Line 14-15 - Are the authors' referencing their own previous work here? If so, this should be made clear: "Recently, we demonstrated that... "

AR2.2: Yes, this is based on previous work. We implemented the suggested wording in the updated manuscript.

RC2.3: Line 17-18 - Suggest being more specific about the types of results being examined e.g. "...why different ESMs produce different spatiotemporal distributions of phytoplankton biomass"

AR2.3: Changed the text in the updated manuscript to the suggested wording.

RC2.4: Line 20 - Three test cases are mentioned, but only two are elaborated in the abstract.

AR2.4: Added additional information to the abstract describing the third case.

RC2.5: Line 28 - Yes, but also by the increasingly prohibitive computational expense of adding complexity and resolution.

AR2.5: Added the additional description in the updated manuscript.

RC2.6: Line 33-35 - This seems like a reasonable metric to vary, particularly when modelling different plankton community structures. It is not necessarily variable as a result of uncertainty.

AR2.6: We were not trying to say that it is varied because of uncertainty, but rather that each of the eight ecosystem models that are used in Laufkötter et al. (2015) use different $Q_{10}$ values for the various ways they try to represent it. For example, in Table 3 of Laufkötter et al. (2015), some models use a single value of $Q_{10}$ across temperature and functional groups, while in others it is different over different temperature ranges or across phytoplankton and zooplankton functional groups.

RC2.7: Line 36-49 – This paragraph first mentions the uncertainty associated with ESM input parameters, then the coupled nature of a given input to multiple outputs, and then the difficulty in knowing whether ESMs are modelling the "correct relationships". These are all valid – if separate – points. But I'm struggling to link these points to the proposed 'solution' in lines 47-49. Will the NNE help to identify which relationships are 'most correct', or extract new 'more correct' relationships? Or is its primary function to more clearly identify the reasons why ESM predictions of biological variables diverge?

AR2.7: In the updated manuscript, we introduce the concept of NNEs earlier in the introduction and move the objective of the paper into the same paragraph (originally Line 71-72). Along with the comments from Referee 1, we have also more clearly specified the objective.

RC2.8: Line 56-59 - This definition was a little confusing to read. Are the "intrinsic relationships" those which are known as true drivers of a target variable? E.g. those captured by lab growth rate experiments, or, as in the current context, the biogeochemical equations underlying ESMs?

AR2.8: It mainly depends on the context of the dataset. Intrinsic relationships in the real-world could be things like lab growth rate experiments in which one driver (such as a nutrient) is varied for a single model organism. Intrinsic relationships in the context of ESMs would be the biogeochemical equations that are programmed into them, which generally have a functional form similar to what is observed in laboratory experiments. In both cases, intrinsic relationships refer to the fundamental relationships that are driving a system forward at the smallest timescale for which data is available.

In the updated manuscript, we have revised the description of the intrinsic relationship ESM example.

RC2.9: Line 61-64 - Similar to the previous point, are your "apparent relationships" a reference to data-derived correlations? For the record, I really like the terms "intrinsic" and "apparent", but I think your description of these terms is much more clear in your previous work "Can machine learning extract the mechanisms controlling phytoplankton growth from large-scale observations?".

AR2.9: The terms *intrinsic* and *apparent* relationships are based on the context of the dataset, similar to what we state in AR2.8.

Apparent relationships in the context of ESMs are the relationships that emerge from the output of ESMs where the intrinsic relationships programmed into the model have interacted across time and space and then had their short timescale values averaged into fields, such as monthly averages.

We have added an example of apparent relationships with respect to ESMs in the updated manuscript.

RC2.10: Line 71-72 – Perhaps 'determining *the most significant sources of* differences in ESM outputs'?

AR2.10: Updated with the suggested wording.

RC2.11: Line 73 – Can '*combinations of these two*' be considered as an independent 'primary driver'?

AR2.11: In general, there are two primary drivers that lead to differences in how ESMs simulate phytoplankton biogeography: physical forcings and phytoplankton physiology. Insofar as both of these act to affect nutrient cycling, they can also act in combination to produce indirect impacts.

RC2.12: Line 78-79 - Possibly worth clarifying that you're only referring to the link between circulation changes and patterns of co-limitation in the ESM (to avoid readers' potentially drawing parallels to real ocean dynamics, to which such findings may not apply).

AR2.12: Added a clarification sentence in the updated manuscript.

RC2.13: Line 97 – A reader's question here might be - "why not identical?"

AR2.13: We stated "similar physical circulations" because the physical circulation in our ESM can be slightly affected by the biological cycle by changing the absorption of shortwave radiation. Since we changed the intrinsic biological relationships in Case 2, this results in small differences in the

circulation between the two model runs. We have added a clarification near the referenced section in the updated manuscript.

RC2.14: Line 162 - For all case descriptions, perhaps include details on e.g. how long each ESM was run for (in model years), output formats (e.g. daily/monthly averaged values) and the model resolution.

AR2.14: We have included additional details in the updated manuscript.

RC2.15: Line 166-167 - Perhaps expand on this, as it seems like an important point. We know from your (very clear and helpful!) Fig. 1 that nutrient distribution is coupled to circulation, but biomass itself is not, and that changes in biomass are a function only of nutrient distribution. With this in mind, the reader might be wondering whether it is even possible – given the constraints of BLING - to "push the biology into fundamentally new states" by varying circulation alone.

AR2.15: This is true on short timescales. In short timescales, such as the size of the timestep of BLING (intrinsic relationships), the biology is not being pushed into fundamentally new biological states from changing circulations. However, the apparent relationships arise both from changing combinations of light, macronutrients, micronutrients, and temperature, as well as time averaging of these relationships. There is no guarantee that, for example under climate change, that the primary drivers will combine and average in the same way.

We have updated the definitions and examples of intrinsic and apparent relationships in the updated manuscript.

RC2.16: Line 236-237 - Why were these activation functions chosen?

AR2.16: We chose to use the hyperbolic tangent sigmoid function for the hidden layer because we showed in previous work (Holder and Gnanadesikan 2021; their Appendix B) that the choice of activation function for the hidden layer did not really affect the performance of the NNEs as long as the activation function was non-linear. Specifically, we tested the following activation functions: hyperbolic tangent sigmoid, logarithmic sigmoid, inverse, positive linear, linear, soft maximum, and radial basis. The settings we chose for this current manuscript allowed us to have reasonably fast training times while keeping high performance metrics.

We have reworded the text in the updated manuscript and moved it closer to the beginning of Section 3.4.

RC2.17: Line 257-258 - Perhaps mention these previous sensitivity tests earlier (e.g. line 235+) - "we previously determined that {x,y} were not sensitive to {p,q} (ref) so our individual NN's were constructed using…"

AR2.17: We have moved the text closer to the beginning of Section 3.4 in the updated manuscript.

RC2.18: Line 274-275 - I think it's worth including more detail on the actual data that the NNEs are using. How many datapoints do the training and test sets contain? Are the training and test sets randomly sampled in both time and space from the ESM outputs? Are they drawn from different temporal periods? Or from different spatial regions? Were any resampling techniques employed to address potential imbalance in the randomly-sampled data? Or were the datasets large enough to effectively capture variance? Did you use all of the depth-integrated output data from the model runs for training/testing?

AR2.18: We understand the request for more information on the dataset. This kind of information is a bit of a paradox and largely depends on the preferences of the reviewers. In manuscripts where we put

this information in the first submission, some reviewers request that we put it in a supplement/appendix. In contrast, when we do not include this information in the first submission, some reviewers request this information. It can be difficult to strike a balance between providing too much information and too little information. We have tried to address this here.

In the updated manuscript, we have added Appendix A which contains more specific information on the datasets for each model run in each of the three cases. Appendix A includes information on the size of each dataset, the sizes of the training and testing subsets, and additional information on how the data was partitioned into training and testing subsets. We have also included more information on how the individual NNs were trained in the main body of the text.

With regards to whether the datasets were large enough to capture the variance, the datasets do appear to be large enough to effectively capture variance.

We did not use the depth-integrated data. We only used the surface values since this is where we also have information from remote sensing products in observational datasets. In a forthcoming manuscript, we demonstrate that the analysis developed here can be extended to such products, providing a useful constraint for ESMs.

RC2.19: Line 281-282 – I'd suggest being more explicit here on how r-squared was calculated - this is a notoriously tricky, often misused metric. E.g. NNE predictions of mean annual biomass for each point are plotted against the 'true' ESM values… standard or adjusted, etc.

AR2.19: Yes, $R^2$ can be a tricky metric when used by itself, especially on non-linear models. That is primarily why we also included RMSE as an additional metric, so they could be considered together.

The $R^2$ calculated when we compare the predictions of the NNEs to the "true" values of the ESM is the square of the Pearson correlation coefficient (i.e., standard $R^2$). The NNEs are predicting the monthly value of biomass (not mean annual biomass) and these are compared against the "true" monthly biomass values from the ESM. We have included this extra information in the updated manuscript.

RC2.20: Line 290-294 – What were your criteria for what constituted a significant increase or decrease in RMSE?

AR2.20: We understand the confusion from our use of the word "significant." We changed this to "substantial" in the updated manuscript so that we are not using a statistical term out of context.

RC2.21: Line 320-321 - I suspect that some readers will have questions about the extremely high performance seen here, across both metrics. It would be helpful again to provide more detail on the nature of the training and test datasets, how they were sampled, etc. Is the "mean value of the total biomass" calculated as a total global mean, or the mean for a given point? Across what time period?

AR2.21: We have included more information about the training and testing subsets and the sampling procedure in the updated manuscript. For more specific information, please see our response (AR2.18) where we address this in more detail.

RC2.22: Line 324-328 - Is this unexpected? The fact that "physical circulation would simply act to change the location of where combinations of light and nutrients were found" seems like a given, considering that "biomass is not directly affected by changes in the physical circulation" in BLING. I think that an explicit clarification of the importance of this result, in this context, would be helpful to place here.

AR2.22: It was not necessarily unexpected, but we were also not certain. We wanted to ensure that this result was verifiable, rather than just assuming, in case there were indirect effects we forgot to consider. This result reinforces what we find in the other cases.

RC2.23: Line 483-485 - This is a really interesting result in terms of the importance of including the correct variables in predictive models. In this case, we happen to know all of the variables that affect our target within the system. When applying such models to realworld data, we don't, and it can have significant consequences for predictive accuracy.

AR2.23: The application to real world data is something that we are currently trying to address in a separate manuscript. However, our preliminary results suggest that the inclusion of about 10 to 11 biogeochemical variables do well at predicting climatological phytoplankton biomass values ($R^2$ values between 0.6 to 0.85).

RC3.1: Line 1-2 - The use of "components" in the title is a little broad. Perhaps substitute with "Drivers of Plankton Biogeography"?

AR3.1: Changed "Components" to "Drivers of Plankton Biogeography."

RC3.2: Line 92-94 - Advise being more specific here - what is meant by "push the biology into fundamentally new states"? I liked "produce new patterns of colimitation", as given in line 79.

AR3.2: Changed to "new patterns of co-limitation."

RC3.3: Line 110 – Introduce symbol for phytoplankton biomass (B) in this line

AR3.3: Added symbol $B$ for phytoplankton biomass.

RC3.4: Line 138 - Typo, should read "than the"

AR3.4: Corrected.

RC3.5: Line 142 - Typo "nutrient and temperature"

AR3.5: Corrected.

RC3.6: Line 180 – Is this meant to read 'Section 3.4'?

AR3.6: Yes, it was supposed to be Section 3.4. This has been corrected in the updated manuscript.

RC3.7: Line 311 - Again, I think your original phrase "new patterns of colimitation" is more descriptive and appropriate than "fundamentally new states"

AR3.7: Changed "push the biology into fundamentally new states," to "lead to new patterns of co-limitation."

RC3.8: Fig. 3 and 4 - Agreed, the inclusion of both large and small phytoplankton in Fig 3 makes it difficult to read. Suggest splitting them up

AR3.8: Could you please clarify? Are you suggesting separate figures for small and large phytoplankton, such that we keep Figure 4 with **only** small phytoplankton and change Figure 3 to have **only** large phytoplankton? Or are you agreeing with the current layout with Figure 3 having **both** large and small phytoplankton and Figure 4 having **only** small phytoplankton?

The reason we chose to include both large and small phytoplankton in Figure 3 was so we could visualize the differences between them on the same plot. We did not give large phytoplankton its own figure since the apparent relationships of the large phytoplankton are already easily visible in Figure 3.

---

## Author Response (AR2)

Dear Editor,

As per the feedback for this next revision, we have made the table captions and text more legible. In particular, we did the following:

- Increased the font sizes of the text within each table.
- Made sure the table and figure captions were not overlapping one another.
- Decreased the width of some columns in some of the tables, so the text was more legible.
- Moved the tables and figures to the end of the manuscript so that each table and figure has its own page. This allowed us to more easily incorporate the captions, tables, and figures for the manuscript, rather than trying to grapple with formatting that would place the figures/tables within the body of the manuscript.
- For the "text" document only (not the updated manuscript PDF), we have changed the section with the tables to a landscape layout, instead of portrait. This allowed us to include the full content of each table while ensuring they were completely visible and legible.

We appreciate your consideration of our manuscript submission to GMD.

Thank you,
Christopher Holder